# Optimal Sparsity of Mixture-of-Experts Language Models for Reasoning Tasks

**Taishi Nakamura**[1,2]**, Satoki Ishikawa**[1]**, Masaki Kawamura**[1]**, Takumi Okamoto**[1,2]
**Daisuke Nohara**[1]**, Jun Suzuki**[3,4,2]**, Rio Yokota**[1,2]
[1]Institute of Science Tokyo, [2]NII LLMC, [3]Tohoku University, [4]RIKEN
`{taishi,rioyokota}@rio.scrc.iir.isct.ac.jp`

## Abstract

Empirical scaling laws have driven the evolution of large language models (LLMs), yet their coefficients shift whenever the model architecture or data pipeline changes. Mixture-of-Experts (MoE) models, now standard in state-of-the-art systems, introduce a new sparsity dimension that current dense-model frontiers overlook. We investigate how MoE sparsity influences two distinct capability regimes: memorization skills and reasoning skills. By training MoE families that vary total parameters, active parameters, and top-$k$ routing under fixed compute budgets, we disentangle pre-training loss from downstream accuracy. Our results reveal two principles. First, **Active FLOPs**: models with identical training loss but greater active compute achieve higher reasoning accuracy. Second, **Total tokens per parameter (TPP)**: memorization tasks improve with more parameters, while reasoning tasks benefit from optimal TPP, indicating that reasoning is data-hungry. Neither reinforcement learning post-training (GRPO) nor increased test-time compute alters these trends. We therefore argue that optimal MoE sparsity must be determined jointly by active FLOPs and TPP, revising the classical picture of compute-optimal scaling. Our model checkpoints, code and logs are open-source at `https://github.com/rioyokotalab/optimal-sparsity`.

## 1 Introduction

The recent evolution of large language models (LLMs) has been driven by empirical scaling laws (Hestness et al., 2017) that link training loss to model size, dataset size, and compute budget. Kaplan et al. (2020) showed that these laws hold across seven orders of magnitude, establishing them as a reliable extrapolation tool for dense Transformers. Subsequent work by Hoffmann et al. (2022) demonstrated that scaling curves can be inverted to choose the compute-optimal combination of parameters and tokens for a fixed budget. Together, these results have made scaling analysis a cornerstone of model planning at both academic and industrial labs.

Yet the coefficients of the scaling laws are not universal. Highly expressive models trained under different optimizers or architectures often follow the same loss trajectory but diverge substantially on downstream reasoning benchmarks (Liu et al., 2023a). Brandfonbrener et al. (2025) extend the classic laws with loss-to-loss prediction, showing that the mapping between training and test distributions admits its own power law when the distributions differ substantially. These observations imply that optimal budgets must be re-estimated whenever we modify the model or the data pipeline.

A particularly compelling architectural modification is the Mixture-of-Experts (MoE) paradigm, offering high capacity at fixed FLOPs by routing each token through a sparse subset of experts (Shazeer et al., 2017; Lepikhin et al., 2021; Fedus et al., 2021). Modern flagship models, e.g., Gemini 2.5 Pro (Gemini Team, 2025), DeepSeek-V3 (DeepSeek-AI, 2025b), and Qwen3 (Qwen Team, 2025) now rely on MoE as a de facto standard for economical scaling. Abnar et al. (2025) derive a parameters-vs-FLOPs frontier and locate an optimal sparsity for a given compute budget. These findings emphasize that the classical dense-model frontier is an incomplete picture, and one must account for architectural knobs such as MoE sparsity and top-$k$ routing.

Evaluating reasoning performance immediately after pre-training overlooks both the benefits of post-training adaptation and the leverage of additional test-time compute. Post-training methods

such as GRPO, which use reinforcement signals to encourage coherent chain-of-thought generation, sharpen a model's reasoning on complex tasks (OpenAI, 2024b; DeepSeek-AI, 2025a). Beyond these refinements, models can further improve outputs at test time by adopting calibrated decoding strategies that mirror how humans pause to reconsider difficult problems. These test-time approaches not only boost routine benchmark performance but, when properly tuned, substantially enhance multi-step mathematical reasoning, demonstrating that adaptive computing at test time is a powerful complement to both model scale and post-training adaptation.

In this paper, we aim to identify how the optimal sparsity of MoE changes between memorization skills (TriviaQA, HellaSwag) and reasoning skills (GSM8K, GSM-Plus) tasks. In this work, we use the term dense models to refer to standard Transformers with a single feed-forward network per layer. For MoE models, we define sparsity as $\text{sparsity} = 1 - \frac{\text{Top-}k}{\text{Experts}}$ following the convention that sparsity measures the fraction of inactive parameters. We train families of MoEs varying not only the total vs. active parameters, but also the number of top-$k$ experts. For each model, we measure the loss on the pre-training data, the task loss on the downstream benchmarks, and the accuracy on those benchmarks. This allows us to disentangle the generalization gap between the train vs. test loss, and the gap between loss vs. accuracy. For both memorization and reasoning benchmarks, the train loss decreases monotonically with the total parameters count increases. The task loss and accuracy follow the same monotonic trend as the train loss for memorization benchmarks. In contrast, for reasoning benchmarks, the task loss and accuracy diverge from this monotonic trend as the total parameters increase and training loss decreases. We find that changing the $k$ in top-$k$ routing has a negligible effect if the number of active parameters is kept constant. We also consider classic generalization-gap controls by sweeping the learning rate and initialization, showing that their effects align strikingly with the generalization-gap caused by sparsity. This confirms that the gap between the performance on memorization skills and reasoning skills can be induced not only by the sparsity of the MoE, but also by classical hyperparameters like learning rate and initialization. We further investigate whether applying GRPO or additional test-time compute could recover the degraded reasoning ability of sparser models. Our results show that the gap between memorization and reasoning performance caused by increased sparsity remains unchanged even after GRPO and increased test-time compute. This means that finding the optimal sparsity of the MoE during pre-training is crucial for training a reasoning model under a fixed compute budget.

We characterize these divergences along two key axes: (1) **Active FLOPs** - downstream reasoning performance is not determined by training loss alone, but by the number of active FLOPs at both train and test time; even at identical training loss, models with a larger top-$k$ consistently outperform smaller ones. (2) **Total tokens per parameter (TPP)** - reasoning ability peaks around 20 tokens per parameter, whereas memorization skills are parameter-hungry and improve with lower TPP. Together, these axes define the compute-optimal sparsity for MoE models.

We further demonstrate that neither reinforcement learning post-training (GRPO) nor additional test-time compute eliminates this trade-off, highlighting that pre-training sparsity remains the dominant factor for reasoning ability under fixed budgets.

Together, our findings refine the scaling laws for MoE LLMs: memorization improves with higher sparsity and more experts whereas reasoning requires a careful balance between active FLOPs and data-per-parameter, occasionally favoring denser configurations in high-compute regimes.

We release model checkpoints, code and logs are open-source at `https://github.com/rioyokotalab/optimal-sparsity`.

## 2 BACKGROUND AND RELATED WORK

### 2.1 MIXTURE OF EXPERTS

**MoE Architecture.** Mixture-of-Experts (MoE) networks were introduced by (Jacobs et al., 1991; Jordan & Jacobs, 1994) and later brought to large-scale neural language modeling by Shazeer et al. (2017). Within the Transformer architecture (Vaswani et al., 2017), MoE layers have proven especially effective, scaling to hundreds of billions of parameters while maintaining manageable training costs (Lepikhin et al., 2021; Fedus et al., 2021; Du et al., 2022). In an MoE layer, a learnable router assigns each token to a sparse subset of experts. Let $\mathbf{x} \in \mathbb{R}^{d_h}$ be a token representa-

tion and $\{\mathrm{FFN}(\mathbf{x})_i\}_{i=1}^n$ the $n$ feed-forward experts. For top-$k$ routing, the router produces scores $\mathbf{s} = \mathbf{x}^\top \mathbf{W}_{\mathrm{router}} \in \mathbb{R}^n$, and selects the indices $\mathcal{K}$ of the $k$ largest components, then normalizes them: $g(\mathbf{x})_i = \frac{\exp(s_i)}{\sum_{j \in \mathcal{K}} \exp(s_j)}$ if $i \in \mathcal{K}$ and $g(\mathbf{x})_i = 0$ otherwise.

The layer output is the weighted sum of the chosen experts: $\mathbf{y} = \sum_{i=1}^n g(\mathbf{x})_i \, \mathrm{FFN}(\mathbf{x})_i$. Modern MoE models typically supplement the token-level cross-entropy loss with two auxiliary terms: a load-balancing loss $\mathcal{L}_{\mathrm{LB}}$, which prevents expert collapse (Shazeer et al., 2017), and a router-z loss $\mathcal{L}_{\mathrm{RZ}}$, which penalizes large router logits for better numerical stability and gradient flow (Zoph et al., 2022). The combined training loss is expressed as $\mathcal{L} = \mathcal{L}_{\mathrm{CE}} + \alpha \mathcal{L}_{\mathrm{LB}} + \beta \mathcal{L}_{\mathrm{RZ}}$, where $\alpha$ and $\beta$ are hyperparameters that control the relative importance of each term in the objective function. This formulation is widely used in recent MoE-based language models and remains unchanged throughout the experiments.

## 2.2 Scaling Laws of LLMs

**Scaling Laws for MoE.** Existing scaling laws demonstrate power-law relationships between model performance, parameter count, dataset size, and compute budget (Kaplan et al., 2020; Hoffmann et al., 2022). Scaling laws for MoE models have similarly explored how total parameter count and expert granularity jointly affect scaling behavior (Clark et al., 2022; Ludziejewski et al., 2024). Building on this, Frantar et al. (2024) derived sparsity-aware scaling exponents that bridge dense and sparse regimes, while Abnar et al. (2025) empirically charted the optimal trade-offs between total parameters and FLOPs per token in MoE settings. Furthermore, recent analyses indicate that increasing sparsity itself can directly improve loss, highlighting sparsity as a key dimension in scaling behavior (Kimi Team, 2025).

**Task Loss.** Since the scaling law for next-token prediction loss does not necessarily align with downstream task loss, it may not be reliable for predicting benchmark performance (Grattafiori et al., 2024). Some work has tried to model downstream accuracy with an exponential curve, but accuracy is only predictable when we average over many tasks and carefully choose which ones to include (Gadre et al., 2024). Another line of research instead first quantifies how downstream task loss scales with parameters and data, then converts predicted losses into accuracy estimates, achieving under two points of absolute error for mid-scale models using minimal extra compute (Bhagia et al., 2024). Prior work observes that downstream task loss relates to pre-training loss, where the shifts depend on the minimal achievable losses determined by the intrinsic complexity and distributional mismatch between the pre-training and downstream datasets (Brandfonbrener et al., 2025).

**Skills** Adding MoE experts tends to improve memorization skills more than reasoning skills, motivating new, generalized scaling frameworks that address scaling laws for reasoning performance (Jelassi et al., 2025). They provide a theoretical explanation for this asymmetry using graph neural networks. Recent work shows that larger top-$k$ can improve compositional generalization in MoE models (Zhao et al., 2025a), though such studies do not observe the non-monotonic effect of sparsity on reasoning performance that we identify. Since scaling laws differ across tasks, the optimal scaling strategy may also vary; for example, knowledge-based QA tasks are "capacity-hungry," benefiting more from larger model sizes, whereas code-related tasks are "data-hungry," benefiting more from increased training data (Roberts et al., 2025). Complementing these observations, recent analyses find that models with identical training loss can still exhibit markedly different reasoning performance (GLM-4.5 Team, 2025), highlighting that reasoning skill depends not only on loss but also on how compute and data are allocated.

## 2.3 Post Training and Test-Time Compute (TTC)

Reinforcement Learning (RL) post-training has long been a predominant approach for improving LLMs. Proximal Policy Optimization (PPO) (Schulman et al., 2017) forms the backbone of RLHF pipelines, from the original GPT alignment work (Ouyang et al., 2022) to the GPT-4 family of models (OpenAI, 2024a). More recently, Group Relative Policy Optimization (GRPO) was introduced as a variant of PPO that replaces the value function baseline with a group-relative advantage estimator, thereby improving memory efficiency and stabilizing updates; this approach already powers frontier-

scale systems such as DeepSeek-R1, achieving state-of-the-art results on mathematical-reasoning benchmarks (Shao et al., 2024; DeepSeek-AI, 2025a).

Complementary to these training-time advances, scaling *test-time compute* (TTC) offers an orthogonal approach. TTC denotes accuracy gains obtained *without* updating model parameters, simply by allocating more inference resources, e.g., running longer chains of thought (OpenAI, 2024b; Muennighoff et al., 2025b), sampling larger candidate pools (Li et al., 2022; Wang et al., 2023; Brown et al., 2024; Schaeffer et al., 2025), or performing explicit search-and-verify steps (Lightman et al., 2024; Shinn et al., 2024; Snell et al., 2025; Inoue et al., 2025). Among these, *self-consistency*, repeated sampling with majority-vote aggregation, has emerged as a strong TTC baseline (Wang et al., 2023).

## 3 EXPERIMENTS

In this section, we empirically demonstrate the scaling of downstream task performance through a systematic investigation of memorization and reasoning skills in MoE LLMs.

### 3.1 EXPERIMENTAL SETUP

We use the Mixtral (Jiang et al., 2024) architecture, a Transformer backbone with RMSNorm (Zhang & Sennrich, 2019), SwiGLU activations (Shazeer, 2020), and rotary positional embeddings (Su et al., 2024). Each feed-forward block is a sparsely gated MoE layer, gated by the dropless token-choice top-$k$ routing (Gale et al., 2023). All models use $L = 16$ layers, following Muennighoff et al. (2025a). We sweep three architectural hyperparameters: (i) the model width $d \in \{512, 1024, 2048\}$; (ii) the number of experts per layer $E \in \{8, 16, 32, 64, 128, 256\}$; and (iii) the top-$k$ experts per token $k \in \{2, 4, 8, 16\}$. Each feed-forward network has a hidden dimension of $2d$. When $d = 512$ and $d = 1024$, we train every combination of $E$ and $k$. For $d = 2048$, we limit the search to $E \leq 128$ due to computational resource constraints.

We train with AdamW (Loshchilov & Hutter, 2019) using a peak learning rate of $4 \times 10^{-4}$, a 2k-step linear warm-up followed by cosine decay, and a weight decay of 0.1. Following Xue et al. (2024) and Zoph et al. (2022), we use the load-balancing and router $z$-losses by $10^{-2}$ and $10^{-3}$, respectively.

**Hyperparameter Study.** To isolate optimization effects, we reuse the same 125B-token corpus. For all HP runs, we fix $E = 16$, $k = 2$, and train two widths, $d_{\mathrm{model}} \in \{512, 1024\}$, with the same FFN expansion factor 2. We vary (i) LM-head initialization schemes, (ii) peak learning rate, and (iii) AdamW $\epsilon$. Further implementation and environmental details are deferred to Appendix A.3.

**Pre-training Datasets.** We use a balanced 125B-token mixture consisting of high-quality web text (43B), mathematics corpora (32B), STEM literature and reference (49B), and code (1B). See Appendix A.1 for complete statistics.

**Evaluation Protocol.** We evaluate three capability areas with standard few-shot prompts. Mathematical Reasoning: GSM8K (Cobbe et al., 2021) (4-shot) and GSM-Plus (Li et al., 2024) (5-shot CoT). Reading Comprehension: TriviaQA (Joshi et al., 2017) with 4-shot prompting. Commonsense Reasoning: HellaSwag (Zellers et al., 2019), each under a 4-shot prompting setup. See Table 3 in Appendix for further details.

### 3.2 DOWNSTREAM PERFORMANCE DOES NOT NECESSARILY IMPROVE WITH TOTAL PARAMETER SIZE

In this section, we examine how the expert sparsity in MoE models affects the relationship between pre-training loss and downstream performance. We train a series of models with controlled sparsity levels and measure their performance on the representative downstream tasks. Our analysis shows that while increasing the total number of parameters reduces pre-training loss, downstream task loss on mathematical reasoning worsens beyond a certain model size.

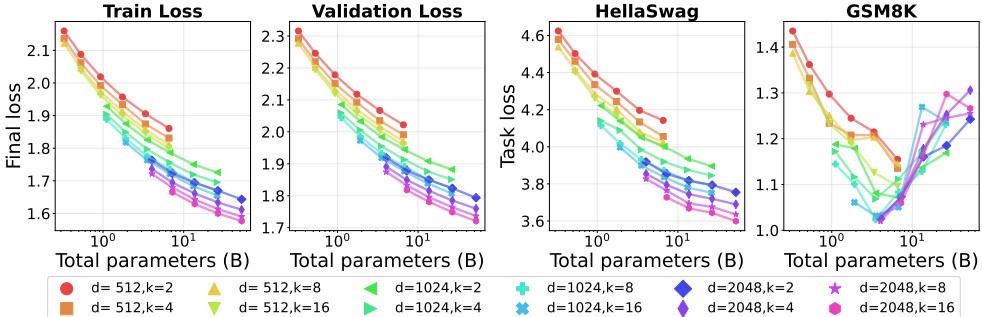

Figure 1: **Although training and validation loss decrease as the total number of parameters grows, the task loss on GSM8K can sometimes worsen with larger models.** Training and validation losses steadily decrease as total or active parameters increase. The HellaSwag task loss follows this scaling trend, whereas GSM8K task loss worsens once total parameters exceed a threshold. Within each fixed top-k group, moving right on the x-axis corresponds to increasing sparsity (because total experts $E$ increases while $k$ remains fixed), so the right-hand task-loss panels implicitly reflect the same sparsity ordering shown explicitly in Figure 5.

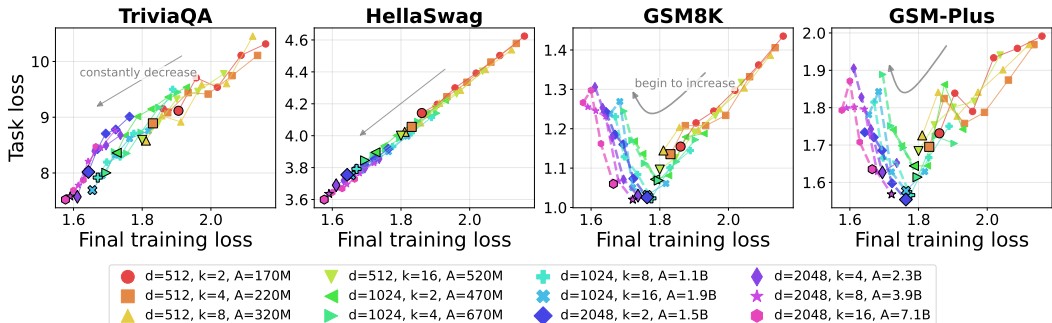

Figure 2: **For GSM8K and GSM-Plus, once the training loss drops below a certain point, the task loss starts to increase.** Results of scaling total parameters by increasing the number of experts, with model width and top-$k$ held constant. For TriviaQA and HellaSwag, the task loss falls monotonically as training loss decreases. By contrast, GSM8K and GSM-Plus show a U-shaped trend: task loss declines with training loss only until a threshold, beyond which further reductions in training loss hurt task performance. That threshold moves lower as active parameter count increases, models with more active parameters achieve a lower optimal task loss. No such active parameters dependence appears for TriviaQA, HellaSwag.

**Task Loss Computation.** Following Brandfonbrener et al. (2025) and Grattafiori et al. (2024), we compute cross-entropy only over the answer tokens by concatenating the prompt with the ground-truth answer. For multiple-choice datasets (e.g., HellaSwag, TriviaQA) the target sequence is the correct answer string, as in Bhagia et al. (2024). For open-ended mathematics datasets such as GSM8K, and GSM-Plus we likewise compute cross-entropy directly against the ground-truth answer tokens.

**Training Loss and Validation Loss.** Figure 1 presents the training and validation losses when fixing the top-$k$, MoE layer width constant and increasing only the number of experts (and hence the total parameter count). As the total parameter count grows, both training and validation losses decrease. Therefore, in terms of pre-training loss, increasing total parameters (thereby raising sparsity) reduces pre-training loss, which is consistent with prior work.

**Experiments with Task Loss** Next, we examine how the downstream task loss responds to increases in the total parameter count. Figure 2 shows task loss on several benchmarks as we vary only the number of experts, holding both top-$k$ and each MoE layer widths constant. On TriviaQA and HellaSwag, lower pre-training loss reduces task loss, indicating that larger total parameter models yield better results on these datasets. In contrast, for GSM8K and GSM-Plus, further reductions

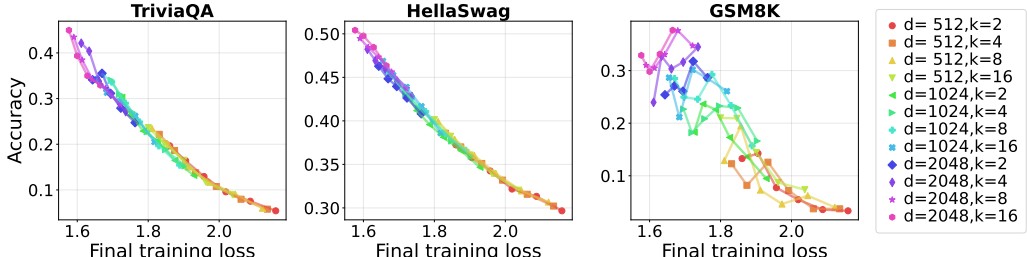

Figure 3: **Downstream accuracy when scaling total parameters via expert count with width and top-$k$ fixed.** TriviaQA and HellaSwag exhibit steadily improving accuracy as pre-training loss decreases, whereas GSM8K shows a non-monotonic trend: further reductions in pre-training loss do not always improve accuracy and can even degrade performance.

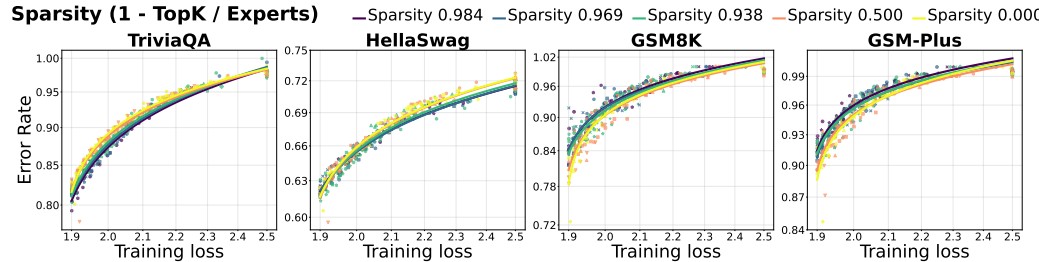

Figure 4: **Effect of sparsity on performance across different tasks** We vary sparsity (1 - top-$k$/Experts) and plot the relationship between pre-training loss and benchmark error rate, including intermediate checkpoints. For TriviaQA and HellaSwag, the error rate clearly tracks training loss and is largely insensitive to sparsity. In contrast, reasoning skills exhibit a strong dependence of error rate on sparsity.

in pre-training loss do not translate into improved task loss; in some cases, the task loss actually worsens. These results suggest that, once top-$k$ and layer width are fixed, an optimal number of experts exists for each task, and adding more beyond that point can harm performance on GSM8K and GSM-Plus.

**Dependence on Active Parameter.** Can we avoid a decline in performance as the total number of experts increases? Figure 2 shows that models with more active parameters begin to overfit at a lower pre-training loss and reach a lower minimum task loss at their optimal expert counts. Consequently, improving results on GSM8K and GSM-Plus requires tuning not only the total number of experts but also the top-$k$ size.

**Downstream Accuracy.** The decline in math-task performance as total parameters increase is not limited to task loss; it also consistently holds for downstream accuracy (Figure 3). For TriviaQA and HellaSwag, accuracy improves monotonically as training loss decreases. By contrast, on GSM8K, further reductions in pre-training loss do not always translate to higher accuracy. When the number of active parameters is held constant, over-optimizing pre-training loss can indeed harm performance. Figure 4 plots benchmark error rate against pre-training loss, including intermediate checkpoints. We observe a sparsity dependence for reasoning skills. These results suggest that, for MoE models, downstream accuracy can deviate from the predictions of conventional scaling laws, and these deviations may vary across different tasks. This effect persists even when controlling for optimization hyperparameters such as learning rate (Appendix C.7).

### 3.3 OPTIMAL SPARSITY FOR ISO-FLOP BUDGETS

We next analyze model quality under a constant compute budget, that is, along *IsoFLOP* contours (Hoffmann et al., 2022; Abnar et al., 2025). For a fixed per-token FLOP count, we vary only the *sparsity configuration*: the number of experts $E$ and the top-$k$ value, while holding the hidden dimension and sequence length.

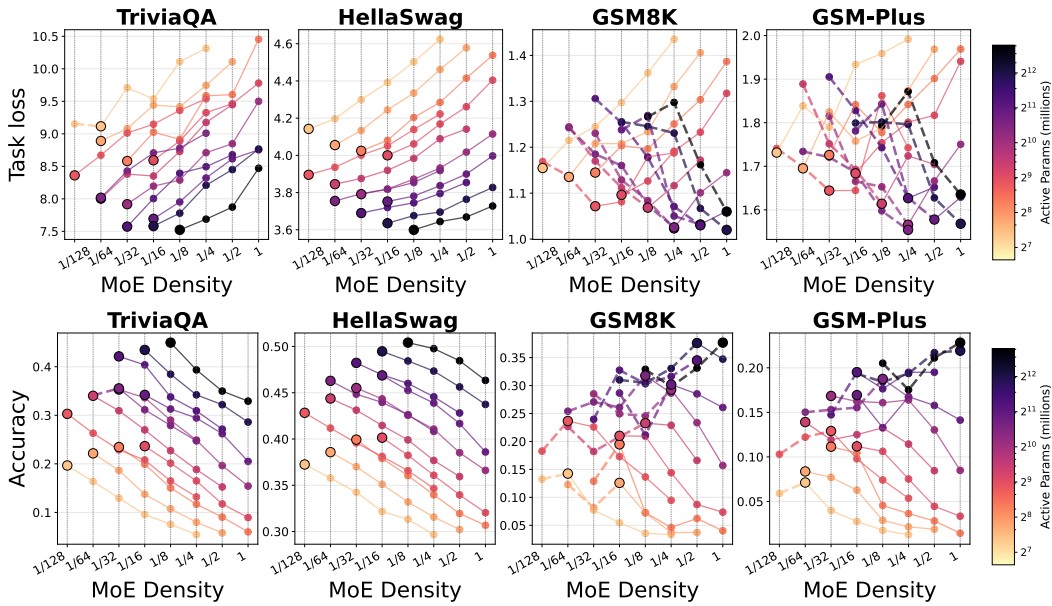

Figure 5: **At fixed active parameter counts, higher sparsity (lower density) consistently improves performance, but at larger active parameter counts, GSM8K and GSM-Plus shift their optima back toward dense models.** Task loss (top row) and Accuracy (bottom row) against the ratio of active experts $k$ to total experts $E$ for a fixed active parameter budget. In the left two tasks (TriviaQA, HellaSwag), increasing sparsity consistently lowers task loss and raises accuracy across all active parameter budgets, in contrast, in the right two tasks (GSM8K, GSM-Plus), once active parameter counts become large, this trend reverses and denser models begin to outperform their sparser counterparts. Dashed segments mark the inverse-scaling regime that starts at the black circle; solid segments show the standard scaling region to the right.

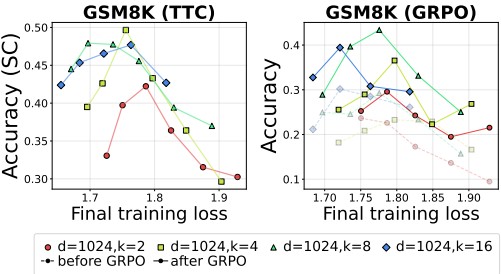

Figure 6: **Effect of Test-Time Compute and GRPO on the loss–accuracy trade-off.** Although both methods yield performance improvements that scale with model size, the loss–accuracy trade-off on GSM8K remains. Left: Final training loss vs accuracy under Test-Time Compute (Self-Consistency). Right: Final training loss vs accuracy after GRPO post-training.

In Figure 5, we show the task-specific optimal sparsity (i.e. 1-TopK/Experts) against model performance under a fixed FLOPs budget. Because the dataset size is fixed in our experiments, increasing the number of active parameters directly increases the training FLOPs. Thus, as the active parameter count grows, the plots implicitly trace how model performance changes with increasing FLOPs. For memorization benchmarks, lower density (higher sparsity) consistently yields lower task loss and higher accuracy. This pattern aligns with prior studies showing that, when FLOPs are fixed to be constant, sparse models outperform denser models on QA tasks (Abnar et al., 2025). By contrast, on mathematical-reasoning benchmarks, denser models outperform their sparser counterparts. At lower FLOPs, increasing sparsity still reduces loss and improves accuracy; however, once the FLOPs budget grows, denser models begin to perform better, achieving both lower loss and higher accuracy. This shift indicates that reasoning skills admit a compute-dependent optimum, rather than monotonically favoring either sparsity or density.

## 3.4 TOKENS PER PARAMETER

The Chinchilla scaling law (Hoffmann et al., 2022) establishes that, under a fixed compute budget, the optimal trade-off between model parameters and training tokens corresponds to approximately

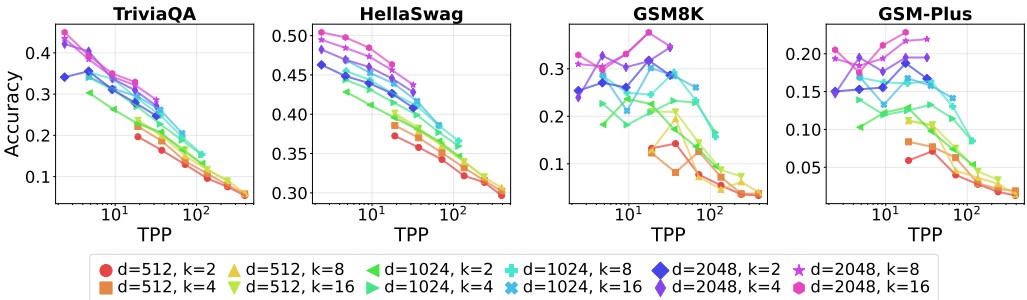

Figure 7: **Effect of TPP on performance across different tasks.** For TriviaQA and HellaSwag, performance improves as the number of parameters increases. In contrast, for reasoning skills, performance deteriorates when the number of parameters becomes too large, indicating that there exists an optimal total tokens per paramete ratio for these tasks. Even at fixed TPP, models with larger top-$k$ values consistently outperform those with smaller top-$k$ on reasoning tasks.

20 tokens per parameter (TPP) for dense models. More recently, Roberts et al. (2025) refined this view by showing that the optimal TPP ratio is task-dependent: memorization skills benefit from lower TPP (i.e., more parameters), whereas reasoning skills benefit from higher TPP (i.e., more data). These findings highlight that TPP should be interpreted not as a universal constant, but as a task-sensitive scaling variable.

In our study, although we varied the number of experts while keeping the total FLOPs fixed, this implicitly altered the TPP measured with respect to total parameters. As shown in Figure 7, this variation reveals distinct behaviors across task categories. For memorization skills, performance improves monotonically as TPP decreases, consistent with the "parameter-hungry" characterization reported by Roberts et al. (2025). For reasoning skills, we observe a non-monotonic trend: accuracy peaks near TPP $\approx$ 20, and degrades when TPP is either too low-when models have too many parameters relative to tokens-or too high—when models have too few parameters relative to tokens.

Furthermore, our experiments reveal that active compute operationalized through the number of top-$k$ experts interacts strongly with TPP. Even at fixed TPP, models with larger top-$k$ values consistently outperform those with smaller top-$k$ on reasoning tasks. This indicates that, in MoE models, reasoning ability depends not only on the Total TPP but also on the balance between total and active parameters. In other words, the discussion of compute-optimal scaling in MoE architectures must explicitly consider both total parameter count and the number of activated parameters per token. We further note that depth ablations (Appendix C.6) exhibit the same non-monotonic dependence on TPP.

## 3.5 IMPACT OF TTC AND POST-TRAINING ON DOWNSTREAM PERFORMANCE

Test-Time Compute and RL post-training are standard for boosting reasoning on tasks such as mathematical problem solving. We therefore investigated whether the performance trade-offs observed above persist or shift when applying (a) Test-Time Compute (TTC) and (b) RL post-training (GRPO). In Test-Time Compute, we evaluated GSM8K(Cobbe et al., 2021) in a zero-shot setting using Self-Consistency (SC) decoding(Wang et al., 2023), generating $2^7$ independent continuations per problem and selecting the most frequent answer. In Post-Training, we fine-tuned each model on the GSM8K training dataset using the GRPO algorithm (Shao et al., 2024). We followed the settings of Zhao et al. (2025b) including reward function and fixed the learning rate constant across all model configurations. Further details regarding the training setup and hyperparameters are provided in Appendix A.2.

As illustrated in Figure 6, neither Test-Time Compute nor GRPO mitigates the GSM8K performance drop that arises when total parameters increase. In other words, although both methods consistently improve overall performance, they do not eliminate the inverted U-shaped relationship between training loss and task accuracy.

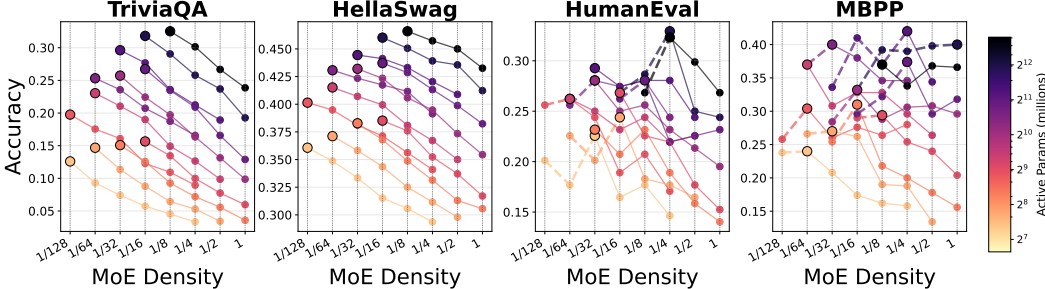

Figure 8: **At fixed active parameter counts, higher sparsity (lower density) consistently improves performance, but at larger active parameter counts, HumanEval and MBPP shift their optima back toward dense models.** Accuracy against the ratio of active experts $k$ to total experts $E$ for a fixed active parameter budget. In the left two tasks (TriviaQA, HellaSwag), increasing sparsity consistently raises accuracy across all active parameter budgets, in contrast, in the right two tasks (HumanEval, MBPP), once active parameter counts become large, this trend reverses and denser models begin to outperform their sparser counterparts. Dashed segments mark the inverse-scaling regime that starts at the black circle; solid segments show the standard scaling region to the right.

## 3.6 CODING TASK ABLATIONS

We evaluate whether the sparsity performance trade offs observed for mathematical reasoning transfer to code generation. Unless otherwise noted, we reuse the same architecture and optimization hyperparameters as in the experimental setup. Models are trained on a 125B token corpus composed of 95B tokens from Stack-Edu Python (Allal et al., 2025) (high-quality educational Python code trained for four epochs following Muennighoff et al. (2023)) and 30B tokens from DCLM-dedup web text (Zyphra, 2024). We assess pass@1 accuracy on HumanEval (Chen et al., 2021) and MBPP (Austin et al., 2021). See Table 3 in Appendix for further details.

Figure 8 summarizes performance as a function of MoE density under matched active compute budgets. As active parameters grow, both HumanEval and MBPP exhibit a clear shift in their optima toward denser configurations: beyond a task-dependent threshold, further increasing sparsity degrades pass@1 despite continued improvements in pre-training loss. This echoes our math findings: when compute allows large active capacity, denser MoE layers yield better procedural reasoning for code synthesis, whereas sparser layers are more favorable only in the low-compute regime. Detailed results for the coding tasks are provided in Appendix C.5.

## 4 DISCUSSION AND LIMITATIONS

All models are trained on a 125B token corpus, which is Chinchilla-optimal for dense models of comparable activated size (Hoffmann et al., 2022). Except for the widest $d = 2048$, top-16 setting, our runs are in or near the Chinchilla-optimal regime, or fall into the overtraining regime. Recent frontier models typically train with a fixed token budget (Grattafiori et al., 2024; Qwen Team, 2025), so holding tokens fixed in our experiments is appropriate. Nevertheless, with larger training corpora, one could train larger models at higher tokens-per-parameter, potentially shifting the optimal sparsity toward sparser configurations, even for reasoning skills. We leave this as a direction for future work.

Our study does not attempt to exhaustively explore all architectural and training settings. The design space of MoE models is vast and it is infeasible to cover every possible combination. Regarding architectural choices, we adopted the Mixtral architecture (Jiang et al., 2024) to ensure comparability with standard dense baselines such as Llama (Touvron et al., 2023a;b; Grattafiori et al., 2024). The Mixtral architecture differs from recent state-of-the-art models like Qwen3 MoE (Qwen Team, 2025) primarily in the use of QK-norm however, since we did not observe training instability, QK-norm was not required in our setting. We also excluded shared experts, as prior work (Muennighoff et al., 2025a) reports mixed or negative results, and our preliminary tests indicated no meaningful performance changes when active and total FLOPs were matched, adding them would only complicate interpretation. For auxiliary losses such as load-balancing and router z-losses, we followed Muennighoff et al. (2025a). Instead, we focused on a representative but systematic sweep over width,

expert count, and top-$k$ routing, which already reveals new regularities that prior work did not capture. In particular, we identify an inverted-U relationship between sparsity and reasoning performance that contradicts the monotonic trends often assumed in scaling analyses. Intuitively, this inverted-U behavior can be explained by two compute-related factors. First, reasoning tasks require substantially higher active compute (i.e., inference FLOPs). Prior work on test-time compute by Snell et al. (2025) demonstrates that increasing compute directly improves reasoning performance. In MoE models, raising the top-$k$ increases the number of experts contributing to each token and thereby increases available inference compute. Second, reasoning requires high training intensity per parameter. Under a fixed training budget, increasing sparsity spreads tokens across more experts, leaving each expert data-starved. While this increased capacity benefits memorization by enlarging the model's effective storage, reasoning becomes either data-starved (when TPP is too low) or under-parameterized (when TPP is too high). These intuitive mechanisms align with the empirical TPP patterns analyzed in Section 3.4.

## 5 CONCLUSION

We investigated the optimal sparsity of MoE language models through a large-scale exploration of Mixtral-style architectures, varying expert count, top-$k$ routing, and width, and evaluating across pre-training, RL post-training, and test-time compute. Our results reveal two central insights. First, **FLOPs matter**: downstream reasoning quality is determined not by pre-training loss alone, but by the number of active FLOPs at both train and test time. Larger top-$k$ consistently outperforms smaller ones even when pre-training loss is matched. Second, **TPP matters**: the tokens-per-parameter ratio governs task-specific scaling. Memorization tasks are parameter-hungry and benefit from sparsity, while reasoning tasks are data-hungry and peak near 20 tokens per parameter, degrading when TPP becomes too low. Together, these findings refine current scaling practice. Sparsity and more experts improve memorization under fixed budgets, but reasoning requires balancing active FLOPs with TPP. In high-compute regimes, the optimal density depends jointly on compute and data: with limited data, denser MoE layers preserve reasoning, while with abundant data, greater sparsity remains advantageous.

## 6 REPRODUCIBILITY STATEMENT

Due to the anonymity requirements and the design of OpenReview, at the time of ICLR submission, the following resources are made available to the reviewers. All source codes, including those for MoE training and evaluation are provided in the supplementary material. The training data used in this study is publicly available. The model checkpoints are not shared due to their large file sizes, which makes anonymous sharing infeasible. Similarly, while we plan to release the training logs via wandb, maintaining anonymity remains a challenge, so they are not included at this stage.

The training setup is described in Section 3.1 and Appendix A, and the evaluation methodology is provided in Appendix B.

## ACKNOWLEDGEMENTS

This work was supported by the "R&D Hub Aimed at Ensuring Transparency and Reliability of Generative AI Models" project of the Ministry of Education, Culture, Sports, Science and Technology. We used ABCI 3.0 provided by AIST and AIST Solutions with support from "ABCI 3.0 Development Acceleration Use. This work used computational resources TSUBAME4.0 supercomputer provided by Institute of Science Tokyo through the HPCI System Research Project (Project ID: hp260015). The authors would like to thank Yukito Tajima for providing valuable and insightful discussions and feedback during the preparation of this manuscript.

## AUTHOR CONTRIBUTIONS

Taishi Nakamura prepared the pretraining datasets, conducted all pre-training experiments and evaluations (excluding test-time-compute), and co-designed the overall experimental setup. Satoki

Ishikawa co-designed the experiments and formulated the overall research strategy. Masaki Kawamura initiated the post-training and test-time-compute (TTC) experiments. Takumi Okamoto conducted the post-training experiments and carried out the Max-Eigen (linear-layer) experiments. Daisuke Nohara conducted TTC experiments. Rio Yokota and Jun Suzuki provided guidance and oversight throughout the project. All authors contributed to manuscript writing and approved the final version.

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

Table 1: Breakdown of the 125 B-token pre-training corpus.

| Source | Type | Tokens | Corpus | Hugging Face or GitLab |
|---|---|---|---|---|
| **High Quality Web** | | | | |
| DCLM-Deduped | High quality web | 33.5B | 788.5B | Zyphra/dclm-dedup |
| Flan decontaminated | High quality web | 9.2B | 18.5B | allenai/dolmino-mix-1124 |
| WebInstructFull | High quality web | 14.7M | 29.7M | TIGER-Lab/WebInstructFull |
| **STEM Literature & Reference** | | | | |
| peS2o | Academic papers | 31.1B | 62.9B | allenai/dolma |
| ArXiv | STEM papers | 11.0B | 22.2B | allenai/dolma |
| Wikipedia | Encyclopedic | 2.3B | 4.7B | gitlab.llm-jp.nii.ac.jp/datasets/llm-jp-corpus-v3 |
| Wikipedia & Wikibooks | Encyclopedic | 1.9B | 3.9B | allenai/dolma |
| Project Gutenberg | Books | 2.7B | 5.5B | allenai/dolma |
| **Mathematics** | | | | |
| OpenWebMath | Math | 6.6B | 13.4B | allenai/dolma |
| Algebraic Stack | Math | 6.6B | 13.3B | allenai/dolma |
| FineMath-4+ | Math | 5.1B | 10.3B | HuggingFaceTB/finemath |
| MathPile commercial subset train split | Math | 4.5B | 9.2B | GAIR/MathPile_Commercial |
| TinyGSM-MIND | Synthetic math | 3.4B | 6.9B | allenai/olmo-mix-1124 |
| OpenMathInstruct-2 | Synthetic math | 2.6B | 5.2B | nvidia/OpenMathInstruct-2 |
| MathCoder2 Synthetic | Synthetic Math | 2.0B | 4.1B | allenai/olmo-mix-1124 |
| StackMathQA | Math | 529.6M | 1070.0M | math-ai/StackMathQA |
| NaturalReasoning | General reasoning | 506.0M | 1022.2M | facebook/natural_reasoning |
| NuminaMath-CoT train split | CoT reasoning | 221.0M | 446.4M | AI-MO/NuminaMath-CoT |
| OpenMathInstruct-1 train split | Synthetic math | 168.4M | 340.2M | nvidia/OpenMathInstruct-1 |
| TuluMath | Synthetic math | 123.9M | 250.4M | allenai/olmo-mix-1124 |
| Metamath OWM-filtered | Math | 42.3M | 85.4M | allenai/olmo-mix-1124 |
| Orca-Math | Synthetic math | 33.5M | 67.7M | microsoft/orca-math-word-problems-200k |
| Dolmino SynthMath | Synthetic math | 15.7M | 31.7M | allenai/olmo-mix-1124 |
| GSM8K train split | Math | 1.4M | 2.8M | allenai/dolmino-mix-1124 |
| GSM8K train split | Math | 1.4M | 2.8M | openai/gsm8k |
| CodeSearchNet-owmfilter | Math | 1.1M | 2.2M | allenai/dolmino-mix-1124 |
| **Code** | | | | |
| StackExchange | CodeText | 725.1M | 1464.8M | allenai/dolmino-mix-1124 |
| **Grand total** | | **125.0B** | **973.4B** | |

## A  Training Setup

### A.1  Pre-training Dataset Details

Table 1 details the pre-training corpus: for each subset, it lists the Hugging Face repository, split identifier, and public URL, alongside the original size and the number of subsampled tokens we used (125 B tokens in the 99:1 train/validation split, as counted by the llm-jp tokenizer v3 with 99,487 tokens). Thus, the total token budget is fixed in strict accordance with Kaplan's scaling law (Kaplan et al., 2020), meaning the observed loss increase (and the accompanying puzzling overfitting that mirrors behavior recently reported by (OLMo, 2025; OpenAI, 2024a)) cannot be attributed to any change in data volume.

### A.2  Post-Training Details

We use GRPO(Shao et al., 2024) with a batch size of 1024, train for 15 epochs totaling 105 steps, and truncate prompts and generated sequences to 512 and 1024 tokens respectively. The actor's learning rate is fixed at $5 \times 10^{-6}$; the temperature is set to 1.0, the KL-penalty coefficient to $10^{-3}$, and 5 samples are used per prompt. Optimisation employs Adam with $\beta = (0.9, 0.999)$, $\epsilon = 10^{-8}$, and weight decay of $10^{-2}$. Following Zhao et al. (2025b), we implemented a code-execution-based evaluator supporting TinyGSM-style and OpenMathInstruct-1 outputs. For a width of 2048 with 16 or 64 experts, we swept the learning rate (Fig. 9) and subsequently fixed it to $5 \times 10^{-6}$ for all GRPO experiments.

### A.3  Implementation & Training Environment

We executed all pre-training runs on the ABCI 3.0 supercomputer (Takano et al., 2024), equipped with NVIDIA H200 GPUs with board-level power capped at 500 W per GPU. TTC experiments

Table 2: Detailed composition of the 125 B-token pre-training corpus *without* GSM8K and its synthetic variants (used for the ablation in Section C.3). Token counts and raw corpus sizes are listed for each source, following the same category structure as Table 1.

| Source | Type | Tokens | Corpus | Hugging Face or GitLab |
|---|---|---|---|---|
| **High Quality Web** | | | | |
| DCLM-Deduped | High quality web | 33.5B | 788.5B | Zyphra/dclm-dedup |
| Flan decontaminated | High quality web | 9.2B | 18.5B | allenai/dolmino-mix-1124 |
| WebInstructFull | High quality web | 14.7M | 29.7M | TIGER-Lab/WebInstructFull |
| **STEM Literature & Reference** | | | | |
| peS2o | Academic papers | 31.1B | 62.9B | allenai/dolma |
| ArXiv | STEM papers | 11.0B | 22.2B | allenai/dolma |
| Wikipedia | Encyclopedic | 2.3B | 4.7B | gitlab.llm-jp.nii.ac.jp/datasets/llm-jp-corpus-v3 |
| Wikipedia & Wikibooks | Encyclopedic | 1.9B | 3.9B | allenai/dolma |
| Project Gutenberg | Books | 2.7B | 5.5B | allenai/dolma |
| **Mathematics** | | | | |
| OpenWebMath | Math | 8.2B | 13.4B | allenai/dolma |
| Algebraic Stack | Math | 8.1B | 13.3B | allenai/dolma |
| FineMath-4+ | Math | 6.3B | 10.3B | HuggingFaceTB/finemath |
| MathPile commercial subset train split | Math | 5.6B | 9.2B | GAIR/MathPile_Commercial |
| MathCoder2 Synthetic | Synthetic Math | 2.5B | 4.1B | allenai/olmo-mix-1124 |
| StackMathQA | Math | 653.9M | 1070.0M | math-ai/StackMathQA |
| NaturalReasoning | General reasoning | 624.7M | 1022.2M | facebook/natural_reasoning |
| NuminaMath-CoT train split | CoT reasoning | 272.8M | 446.4M | AI-MO/NuminaMath-CoT |
| TuluMath | Synthetic math | 153.0M | 250.4M | allenai/olmo-mix-1124 |
| Metamath OWM-filtered | Math | 52.2M | 85.4M | allenai/olmo-mix-1124 |
| Orca-Math | Synthetic math | 41.4M | 67.7M | microsoft/orca-math-word-problems-200k |
| CodeSearchNet-owmfilter | Math | 1.1M | 2.2M | allenai/dolmino-mix-1124 |
| **Code** | | | | |
| StackExchange | CodeText | 725.1M | 1464.8M | allenai/dolmino-mix-1124 |
| **Grand total** | | **125.0B** | **961.0B** | |

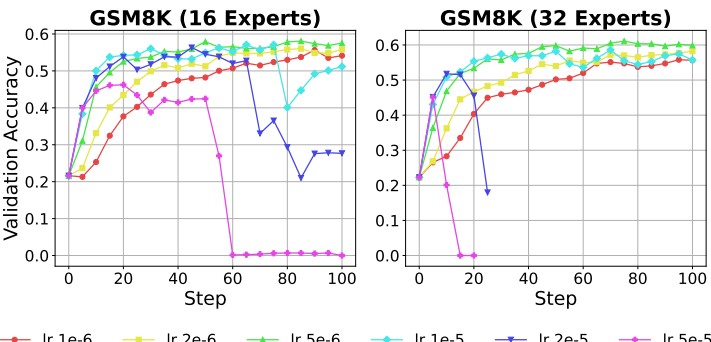

Figure 9: **Learning-rate sweep for width** $= 2048$. We varied the number of experts and swept the learning rate. For both 16 and 32 experts, $5 \times 10^{-6}$ produces the most stable training.

were conducted on the TSUBAME 4.0 supercomputer at the Global Scientific Information and Computing Center, Institute of Science Tokyo. They used NVIDIA H100 SXM5 94 GB GPUs (four GPUs per node) and InfiniBand NDR200 interconnects for inter-node communication.

For pre-training, we extended the Megatron-LM[1] codebase to add functionality needed for this study, with support for pipeline, tensor, and expert parallelism. Reinforcement learning experiments were implemented using GRPO (Shao et al., 2024) on top of the veRL[2] framework. Model quality was assessed using lm-evaluation-harness[3] and LargeLanguageMonkeys[4].

---

[1] https://github.com/NVIDIA/Megatron-LM

[2] https://github.com/volcengine/verl

[3] https://github.com/EleutherAI/lm-evaluation-harness

[4] https://github.com/ScalingIntelligence/large_language_monkeys

Table 3: Evaluation Benchmark Details

| Dataset | TriviaQA | HellaSwag | GSM8K | GSM-Plus | HumanEval | HumanEval+ | MBPP | MBPP+ |
|---|---|---|---|---|---|---|---|---|
| Task | QA | MRC | Math Reasoning | Math Reasoning | Code Reasoning | Code Reasoning | Code Reasoning | Code Reasoning |
| Language | EN | EN | EN | EN | EN | EN | EN | EN |
| # Instances | 17,944 | 10,042 | 1,319 | 10,552 | 164 | 164 | 500 | 378 |
| Few-shot # | 4 | 4 | 4 (0 for TTC) | 5 | 0 | 0 | 3 | 3 |
| Metric | Accuracy | Accuracy | Accuracy | CoT Acc. | Pass@1 | Pass@1 | Pass@1 | Pass@1 |

# B   EVALUATION SETUP

We evaluate our models using the lm-evaluation-harness framework (Gao et al., 2024) across four key capability areas. All evaluations employ standard few-shot prompting strategies unless otherwise specified.

We assess logical reasoning capabilities using Mathematical problem-solving is evaluated using GSM8K (Cobbe et al., 2021) with 4-shot prompting and GSM-Plus (Li et al., 2024) with 5-shot CoT prompting. We evaluate comprehension abilities using TriviaQA (Joshi et al., 2017) with 4-shot prompting. Common sense reasoning is assessed through HellaSwag (Zellers et al., 2019) using 4-shot prompting setups. Finally, code reasoning capabilities are benchmarked on HumanEval (Chen et al., 2021) and HumanEval+ (Liu et al., 2023b) with 0-shot prompting and MBPP (Austin et al., 2021) and MBPP+ (Liu et al., 2023b) with 3-shot prompting, both evaluated using the Pass@1 metric. For Test-Time Compute (TTC) experiments specifically, GSM8K evaluation is conducted under a zero-shot setting. To accommodate the variety of valid answer formats, we extend the strict match patterns provided by the `lm-evaluation-harness` beyond the standard implementation. Our matching criteria accept both the standard GSM8K format (####) and GSM8K-CoT formats prefixed with "The answer is" or "Answer:".

Table 3 provides comprehensive details for all evaluation benchmarks.

# C   ADDITIONAL EXPERIMENTS

## C.1   GRPO

**Training on MATH 500 Dataset**   Following the analysis presented in Section 3.5, the inverted U-shaped relationship between training loss and task accuracy persists even after applying GRPO. To verify that this phenomenon is not due to performing GRPO on the GSM8K dataset, we conducted additional GRPO experiments on the MATH 500 dataset (Lightman et al., 2024). As illustrated in Figure 10, GRPO on the MATH dataset yields consistent results with those obtained on the GSM8K dataset, confirming that this inverted U-shaped relationship is robust across different GRPO training datasets.

## C.2   TEST-TIME COMPUTE

**Evaluation Setup**   We evaluated both GSM8K(Cobbe et al., 2021) in a purely zero-shot setting using Self-Consistency (SC) decoding(Wang et al., 2023), generating $2^7$ independent continuations per problem and selecting the most frequent answer with 128 samples per problem. Specifically, for each prompt we generated up to 1,024 tokens under temperature 0.6 and nucleus sampling ($\text{top-}p = 0.95$), drawing 128 independent continuations and selecting the most frequent answer.

**Zero-shot VS Few-shot**   To set up Test Time Compute appropriately, we investigate how varying the number of prompt shots affected each expert's behavior (Figure 11). Few shot performance is unstable and dropped significantly for models with a small number of experts, so we use zero shot inference for Test Time Compute. When few shot chain of thought is used to standardize answer formats, the provided demonstration steps can be internalized as a fixed reasoning pattern by the model. As a result, the model's inherent inference capabilities may not be fully expressed, and its ability to generalize to novel problems could be hindered (Kojima et al., 2022).

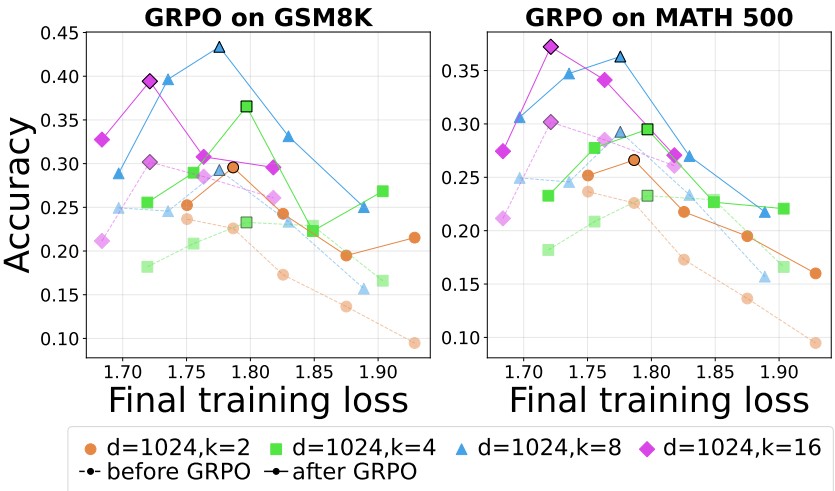

Figure 10: **Comparison of GSM8K accuracy for models fine-tuned with GRPO on different training datasets (left: GSM8K, right: MATH 500).** Performance decline is consistently observed across different training datasets.

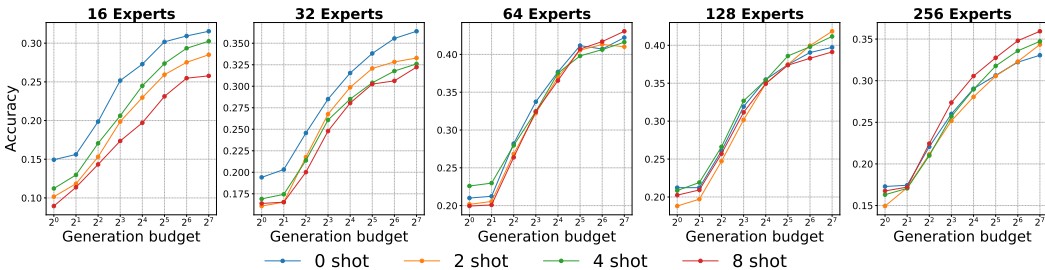

Figure 11: **GSM8K accuracy of model (d=1024) across different shot counts.** Because few shot performance is unstable and dropped significantly for models with a small number of experts, zero shot is used for Test-Time Compute.

**Temperature**    Figure 12 shows that the inverted U-shaped performance-decline trend holds across every temperature setting, indicating that sampling temperature does not affect this behavior. This suggests that, although temperature controls inference randomness, the primary drivers of performance decline are inherent to model architecture rather than temperature settings.

**Evaluation of Larger Generation Budget**    We extended the sample size used for Test-Time Compute as described in Section 3.5, generating a larger set of candidate responses. We then measured the resulting accuracy across different generation budgets to assess how increased sampling influences performance (Figure 13). For an active parameter count of 8 (top-8), the performance decline is gradually mitigated, whereas for an active parameter count of 2 (top-2), the decline is instead amplified, resulting in a more pronounced U-shaped trend. Although increasing the sample count further may provide additional insights, it remains challenging to identify a consistent mitigation pattern across all models.

**Increasing Top-k During Inference**    We compared the performance under TTC for model with a hidden dimension of 2048, 128 experts, and top-2 routing by varying the inference-time top-k parameter. (Figure 14) Specifically, although doubling top-k sometimes yielded temporary improvements in Pass@1, applying TTC ultimately showed that the original top-2 setting maintained the highest performance, suggesting that no fundamental performance gain occurs.

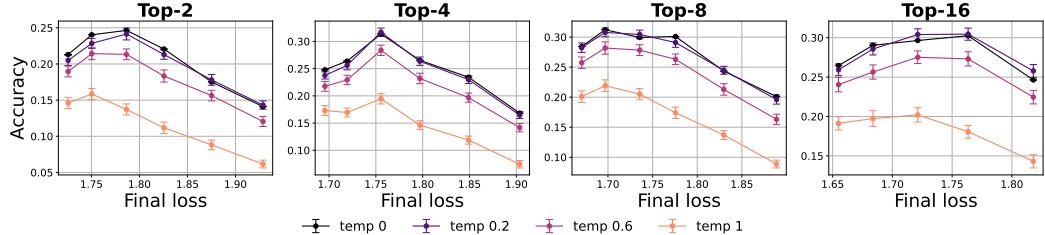

Figure 12: **Comparison of performance decline across different temperature settings (pass@1, d=1024).** A consistent performance decline is observed regardless of temperature, and overall accuracy increases as temperature decreases (i.e., approaches greedy).

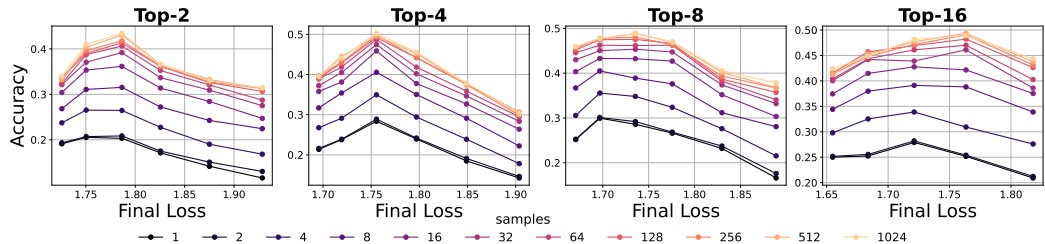

Figure 13: **Accuracy across generation budgets with increased sample counts.** With an active parameter count of 8 (top 8), the performance decline is gradually alleviated as the budget increases, whereas with an active parameter count of 2 (top 2), the decline is amplified, resulting in a more pronounced U shaped trend.

### C.3 GSM8K OVERFITTING ANALYSIS

To investigate whether our model overfits to GSM8K due to the inclusion of GSM8K training data and its synthetic derivatives, we conducted an ablation experiment removing major GSM8K-related datasets from our pre-training corpus as listed in Table 2.

We removed TinyGSM-MIND, both GSM8K train split instances, Dolmino SynthMath, OpenMathInstruct-1, and OpenMathInstruct-2, which contain either the original GSM8K training data or synthetic problems derived from it.

The results are shown in Figure 15 and 16. We observe that the trends with respect to sparsity on GSM8K remain unchanged, both for Pass@1 and TTC metrics. This indicates that while GSM8K training data and its synthetic derivatives do improve GSM8K scores, they do not alter the underlying performance trends. However, after post-training, we observe some changes in these trends, which we leave as future work to investigate further.

### C.4 GSM8K PROBLEM ANALYSIS

We investigated whether models with varying numbers of experts exhibit differences in their ability to solve specific problems on the GSM8K dataset.

Figure 17 shows the results. We observe that different sparsity levels solve different instances of the problems.

### C.5 DETAILED RESULTS FOR CODING TASKS

This appendix provides detailed results for the coding task ablations, mirroring the analyses for mathematical reasoning presented in the main text. Specifically, we detail the non-monotonic relationship between scaling and downstream performance, showing how both task loss (Figure 18, 19) and accuracy (Figure 20) can degrade as pre-training loss improves. We then analyze key ar-

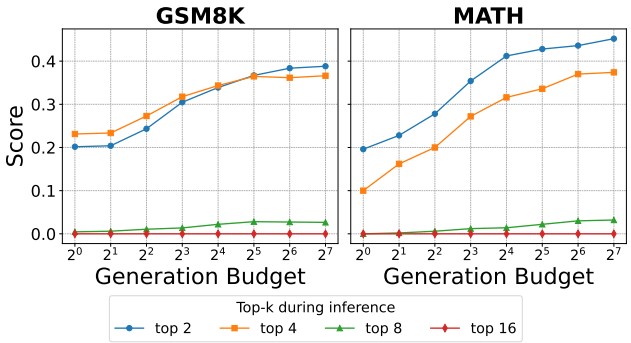

Figure 14: **Increasing the top-k parameter only at inference time does not improve performance.** Performance comparison under TTC for a Mixture-of-Experts model (hidden dimension 2048, 128 experts, top-2) as the top-k parameter is increased. While doubling k can occasionally improve Pass@1, applying TTC ultimately shows that the original top-2 configuration delivers the highest performance.

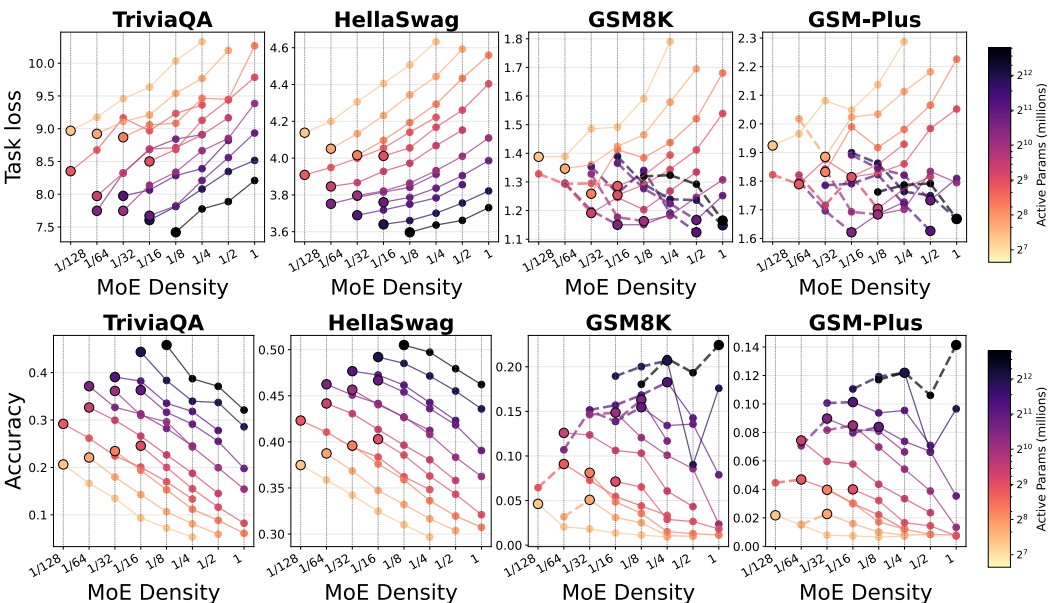

Figure 15: **Performance versus MoE density after removing GSM8K-related training data.** Task loss (top) and accuracy (bottom) are plotted against the ratio of active experts $k$ to total experts $E$ for a fixed active parameter budget. While performance on memorization tasks (TriviaQA, HellaSwag) improves with sparsity, the trend reverses for math reasoning tasks (GSM8K, GSM-Plus) at larger active parameter counts. Dashed segments mark the inverse-scaling regime.

chitectural factors, including the impact of MoE sparsity (Figure 21) and the optimal Tokens-per-Parameter (TPP) ratio (Figure 23). In addition, we report sparsity–accuracy trends for the coding benchmarks HumanEval+ and MBPP+ (Figure 22).

## C.6 ABLATION ON DEPTH

We conducted additional experiments using a 32 layer architecture. Motivated by prior reports suggesting that increased depth can improve performance (Liu et al., 2024; Team, 2024; Ye et al., 2025), we evaluated whether deeper models exhibit similar trends in our setting. For the 32 layer configuration, we observed that the results align with the patterns discussed in the previous section, when analyzed through the lens of TPP, the behavior remains consistent with our earlier findings.

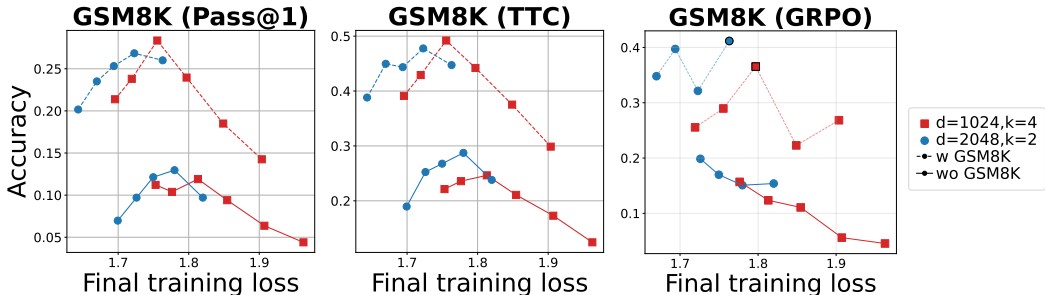

Figure 16: GSM8K performance without GSM8K-related training data: Pass@1 (left), TTC with 128 budget (center), and after GRPO (right)

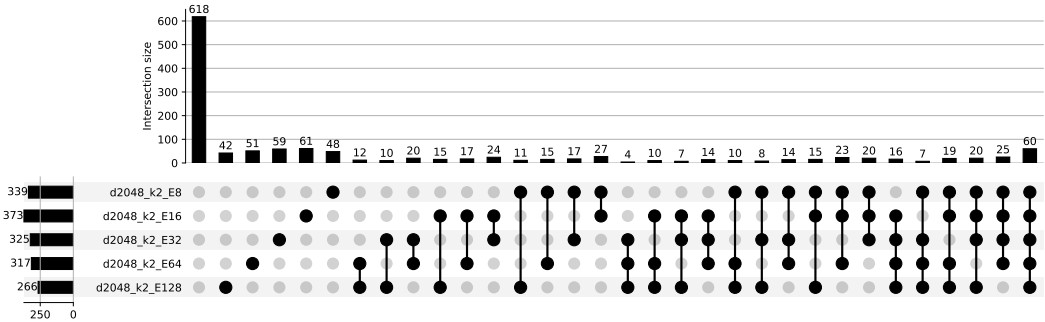

Figure 17: **Analysis of solvable problems across different numbers of experts on GSM8K.** This graph displays the number of problems that were commonly solvable or unsolvable across models with varying numbers of experts.

### C.7 INFLUENCE OF OPTIMIZATION HYPERPARAMETER

Thus far, we have demonstrated that the structure of the model, particularly the degree of sparsity, can lead to differences in reasoning performance on downstream tasks, even when the models converge to the same training loss. Such differences are similar to generalization, in which a model's behavior on unseen data reflects implicit inductive biases rather than mere fit to the training data. Studies on neural network generalization have long recognized that not only architectural choices, but also optimization dynamics (i.e., differences in hyperparameter settings, regularization schemes, and optimizer algorithms), play an important role in shaping these inductive biases. Motivated by this insight, we examine the learning-rate scale, which is critical to generalization (Keskar et al., 2017; Li et al., 2019; Yang & Hu, 2021). Our goal is to investigate how these choices influence the model's ability to transfer to downstream tasks, beyond what is captured by pre-training loss alone.

Figure 25 illustrates our empirical findings, obtained using a MoE architecture with 16 experts. While memorization skills remains largely invariant to these hyperparameters, reasoning skills are sensitive to the learning rates: when models converge to the same training loss, trainings with lower learning rates and smaller initialization scales yield superior downstream accuracy. These observations carry an important implication. Studies on generalization in large-scale language models should incorporate rigorous reasoning benchmarks rather than relying solely on validation loss curves or standard QA tasks to fully capture the impact of optimization-induced implicit biases. This enables a more precise analysis on the generalization of LLMs.

### C.8 ABLATION ON TOKEN BUDGET

To validate our choice of the 125B token budget used in the main experiments, we conducted a preliminary ablation study scaling the training duration to 1T tokens. We compared the baseline model trained on 125B tokens against a model trained on 1T tokens. The training corpus for the

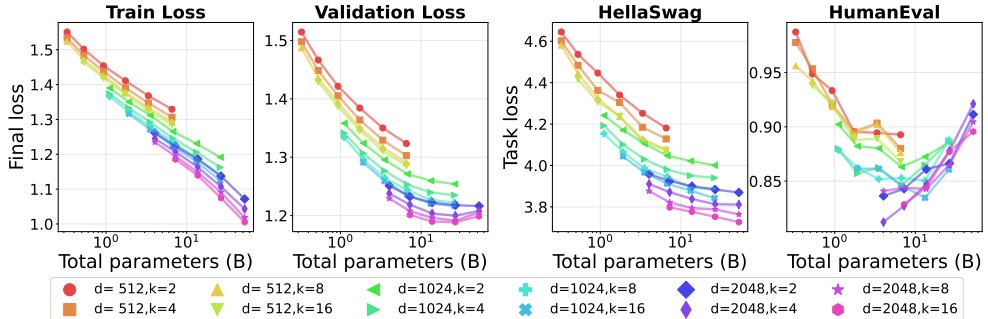

Figure 18: **Although training and validation losses generally decrease as the total number of parameters increases, validation loss for the largest models does not fully converge.** HellaSwag task loss follows this favorable scaling trend, but HumanEval task loss sometimes worsens once the total number of parameters exceeds a certain threshold.

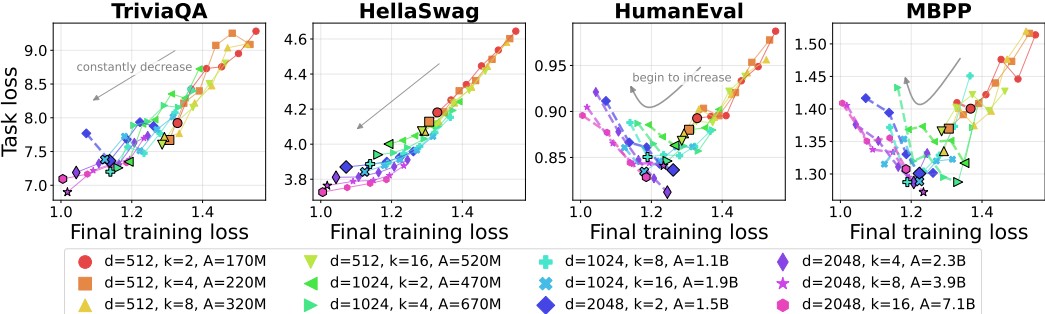

Figure 19: **For HumanEval and MBPP, once the training loss drops below a certain point, the task loss starts to increase.** Results of scaling total parameters by increasing the number of experts, with model width and top-$k$ held constant. For TriviaQA, HellaSwag, and task loss falls monotonically as training loss decreases. By contrast, HumanEval and MBPP show a U-shaped trend: task loss declines with training loss only until a threshold, beyond which further reductions in training loss hurt task performance.

1T run was constructed by upsampling the 973.4B token dataset (Table 1) via random sampling to reach the 1T target. For this experiment, we used a model configuration with width $d = 2048$, 16 experts ($E = 16$), and top-2 routing ($k = 2$). Hyperparameters followed the settings described in Section 3.1, with the exception of the learning rate schedule, where the cosine decay was stretched to accommodate the 1T token duration.

The results are presented in Table 4. Since the extended 1T corpus is predominantly composed of general web text, which serves as a rich source of world knowledge, memorization-intensive tasks such as TriviaQA and HellaSwag exhibited significant performance gains. Conversely, reasoning tasks (GSM8K and GSM-Plus) showed comparatively marginal improvements. We attribute this discrepancy to the scarcity of high-quality mathematics and reasoning corpora available in the open source at the time of experimentation; simply scaling up the volume of general web text contributes minimally to reasoning capabilities. Therefore, extending the training to 1T tokens significantly increases computational costs without yielding proportional gains in reasoning performance, justifying our adoption of the 125B token budget to prioritize examining architectural trade-offs.

# D  THE USE OF LARGE LANGUAGE MODELS

Use of Large Language Models (LLMs). In preparing this paper, we used large language models only for minor editing tasks, such as improving grammar and clarity of exposition. LLMs did not contribute to the research ideas, methodology, experiments, or analysis, and thus played no substantive role in the scientific contributions of this work.

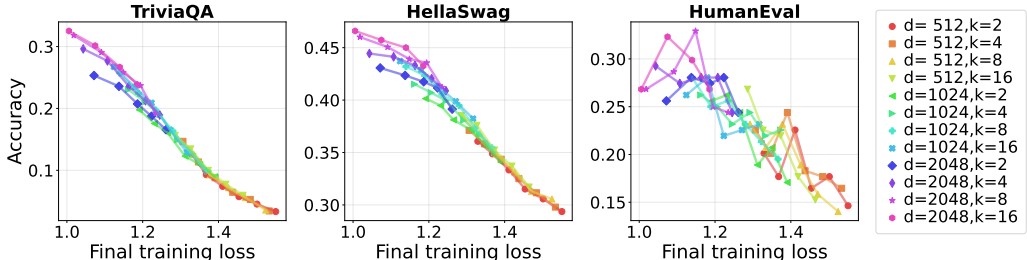

Figure 20: **Downstream accuracy when scaling total parameters via expert count with width and top-$k$ fixed.** TriviaQA and HellaSwag exhibit steadily improving accuracy as pre-training loss decreases, whereas HumanEval shows a non-monotonic trend: further reductions in pre-training loss do not always improve accuracy and can even degrade performance.

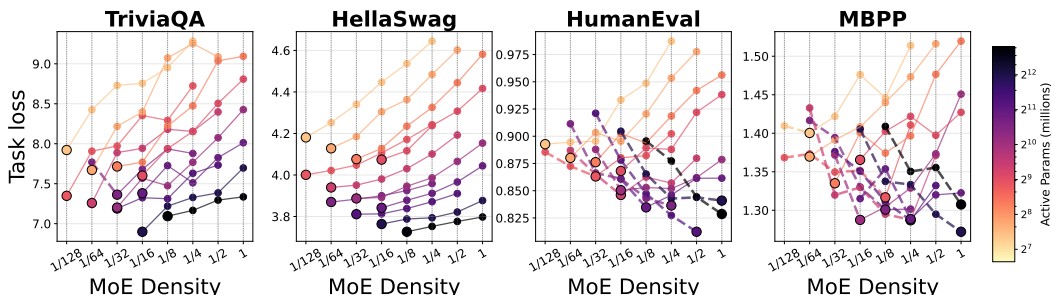

Figure 21: **At fixed active parameter counts, higher sparsity (lower density) consistently improves performance, but at larger active parameter counts, HumanEval and MBPP shift their optima back toward dense models.** Task loss against the ratio of active experts $k$ to total experts $E$ for a fixed active parameter budget. In the left two tasks (TriviaQA, HellaSwag), increasing sparsity consistently lowers task loss across all active parameter budgets, in contrast, in the right two tasks (HumanEval, MBPP), once active parameter counts become large, this trend reverses and denser models begin to outperform their sparser counterparts. Dashed segments mark the inverse-scaling regime that starts at the black circle; solid segments show the standard scaling region to the right.

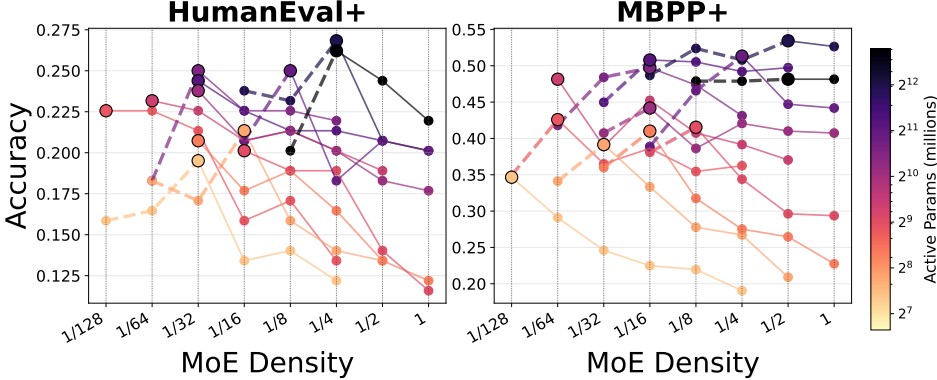

Figure 22: **At fixed active parameter counts, higher sparsity (lower density) consistently improves performance, but at larger active parameter counts, HumanEval+ and MBPP+ shift their optima back toward dense models.** Accuracy against the ratio of active experts $k$ to total experts $E$ for a fixed active parameter budget. Once active parameter counts become large, this trend reverses and denser models begin to outperform their sparser counterparts. Dashed segments mark the inverse-scaling regime that starts at the black circle; solid segments show the standard scaling region to the right.

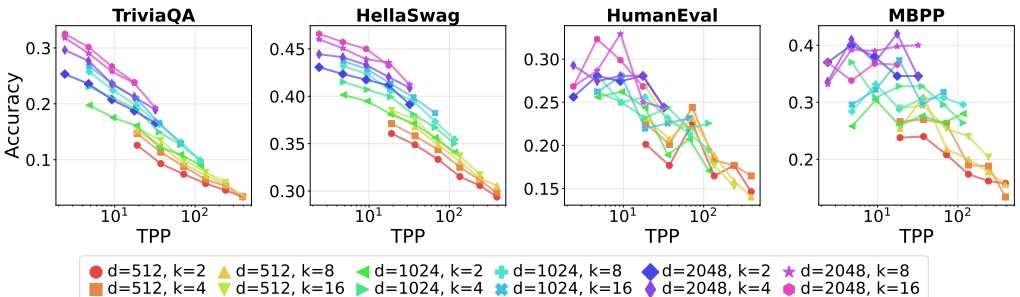

Figure 23: **Effect of TPP on performance across different tasks.** For TriviaQA and HellaSwag, performance improves as the number of parameters increases. In contrast, for reasoning-intensive tasks such as HumanEval and MBPP, performance deteriorates when the number of parameters becomes too large, indicating that there exists an optimal data to parameter ratio for these tasks.

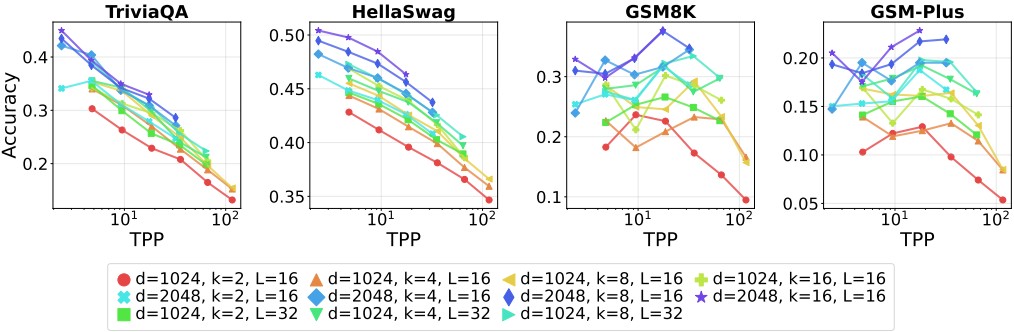

Figure 24: **Effect of model depth on TPP-performance trade-offs.**

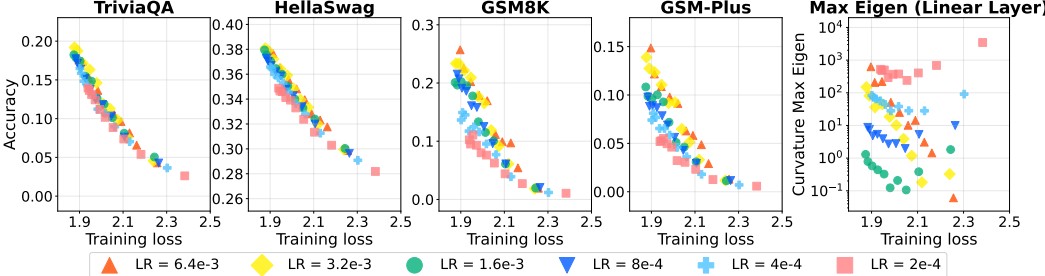

Figure 25: **For reasoning skills, the relationship between training loss and downstream performance is dependent on the choice of optimization hyperparameters.** The learning rate also impacts downstream accuracy. For the maximum eigenvalue, we evaluated the maximum eigenvalue of fisher information matrix under a K-FAC approximation (Martens & Grosse, 2015; Eschenhagen et al., 2023). Following (Grosse et al., 2023), we calculate the maximum eigenvalues only for linear layers. We find that higher learning rates lead to a lower maximum eigenvalue, which is consistent with existing research indicating that convergence to flatter minima improves generalization (Hochreiter & Schmidhuber, 1997; Keskar et al., 2017; Jiang et al., 2020).

Table 4: **Performance comparison between models trained on 125B tokens versus 1T tokens.** The model configuration is fixed at $d = 2048$, $E = 16$, and $k = 2$.

| Token Budget | TriviaQA | HellaSwag | GSM8K | GSM-Plus |
|---|---|---|---|---|
| 125B | 0.279 | 0.426 | 0.318 | 0.188 |
| 1T | 0.535 | 0.538 | 0.363 | 0.214 |

