# OpenReview forum: "Optimal Sparsity of Mixture-of-Experts Language Models for Reasoning Tasks"
_ICLR.cc/2026/Conference — ICLR 2026 Oral_

### Official Review · Reviewer_fnio · 2025-10-23

**Soundness:** 4
**Presentation:** 4
**Contribution:** 3
**Rating:** 8
**Confidence:** 4

**Summary:**

This paper explores how the sparsity of MoE models affects their performance on downstream tasks. While prior scaling law works have mostly focused on pretraining loss or efficiency, this work reveals that downstream memorization and reasoning capabilities respond differently to sparsity. The authors conduct very extensive empirical analysis on a range of benchmarks. Through various analyses, they observe interesting points (see Strengths section) that could be further explored and taken into consideration when building or deploying MoEs.

**Strengths:**

The paper presents several insightful findings based on systematic analysis of MoE sparsity across a variety of experiments.

- Active FLOPs has a significant effect on the downstream task performance, not just determined by pretraining loss alone.
- The study uncovers an important trade-off btw memorization and reasoning with TPP, where memorization skills are parameter-hungry and reasoning skills are data-hungry.
- Post-training or test-time scaling do not change the memorization-reasoning gap, so the optimal sparsity must be determined pretty much during the pretraining stage.

**Weaknesses:**

While the observations are novel and well-supported empirically, I would consider it more significant to find any **intuitive or theoretical rationales** behind them. For instance, why does reasoning capability require denser MoEs, while memorization thrives on sparsity? Why doesn’t simply scaling the parameter count work?

**Questions:**

In Line 321-323:

> *At lower FLOPs, increasing sparsity still reduces loss and improves accuracy; however, once the FLOPs budget grows, denser models begin to perform better, achieving both lower loss and higher accuracy.*

However, I could not find where this trend is clearly demonstrated. Figure 5 appears to fix the FLOPs budget, so it doesn’t reveal how model performance varies as FLOPs increases.

---

> ### Author Response · Authors · 2025-11-24
> **Response to Reviewer 4 (fnio)**
>
> We sincerely thank the reviewer for their thoughtful evaluation and for highlighting the strengths and significance of our analysis of sparsity, memorization, and reasoning in MoE models.
>
> > While the observations are novel and well-supported empirically, I would consider it more significant to find any intuitive or theoretical rationales behind them. For instance, why does reasoning capability require denser MoEs, while memorization thrives on sparsity? Why doesn’t simply scaling the parameter count work?
>
> ## Intuitive and Theoretical
>
> ### Intuitive Perspective
>
> Intuitively, our findings can be explained by two compute-related factors.
> First, reasoning tasks require substantially higher active compute (i.e., inference FLOPs). Prior work on
> test-time compute Snell et al. (2025) demonstrates that increasing compute directly improves reasoning
> performance [1]. In MoE models, raising the top-\(k\) increases the number of experts contributing to each
> token, thereby increasing the available inference FLOPs.
>
> Second, reasoning requires high training intensity per parameter. When the training budget is fixed,
> increasing sparsity distributes tokens across more experts, reducing the number of training tokens each
> expert receives. While this increased capacity benefits memorization where additional parameters
> effectively enlarge the model’s storage capacity, reasoning capabilities become “data-starved.” Our analysis
> (Section 3.4) shows that Tokens-Per-Parameter (TPP) strongly correlates with reasoning performance.
> Moreover, our results indicate that TPP exhibits an optimal regime rather than a monotonic relationship
> with performance. When TPP is too low, experts are data-starved; when TPP is too high, the model
> becomes under-parameterized relative to the available data. This suggests that the optimal sparsity level
> itself depends on the total amount of training data: in regimes with substantially larger token budgets,
> the optimal TPP may shift toward more sparse configurations, as each expert would still receive
> sufficient data even under higher sparsity.
>
> [1] Snell et al. Scaling LLM Test-Time Compute Optimally can be More Effective than Scaling Model Parameters. ICLR ‘25.
>
> ### Theoretical Perspective
>
> Among the prior work summarized in Section 2.2, the study by Jelassi et al. (2025) is the most relevant reference. Using Graph Neural Networks, they provide a theoretical proof that, unlike in memorization tasks, reasoning tasks become less efficient when the model is too sparse, which requires wider MLPs and leading to computational inefficiency. Following the intuition behind their proof, you can interpret this phenomenon as follows: reasoning tasks require integrated information processing across the model's width. In MoE architectures, routing sends each token to only a small number of experts. But reasoning tasks (e.g., shortest paths or connectivity in GNN settings) require combining multiple pieces of information, merging and comparing them, and performing computations over the global structure. With a sparse MoE, tokens are scattered across experts, making it difficult for the necessary information to co-locate; moreover, top-k routing limits cross-expert integration. As a result, the type of "width-wise holistic processing" that dense models perform naturally becomes difficult for sparse models. This provides intuition for why overly sparse MoE architectures tend to underperform on reasoning tasks.
>
> ## Clarification on Figure 5: Scaling Trends
>
> > However, I could not find where this trend is clearly demonstrated. Figure 5 appears to fix the FLOPs budget, so it doesn’t reveal how model performance varies as FLOPs increases.
>
> Thank you for your question. Figure 5 plots performance curves where each connected line represents a fixed active parameter count (Iso-FLOPs).
> In our experimental setup, the dataset size is held constant, which means that the training FLOPs scale directly with the number of active parameters. As a result, increasing the number of active experts increases the total training FLOPs.
> To make this clearer, we have added the following explanation to the revised manuscript:
> “Because the dataset size is fixed in our experiments, increasing the number of active parameters directly increases the training FLOPs.
> Therefore, the trend of performance scaling with FLOPs can be observed by comparing across different active parameter counts (i.e., different lines).

---

> > ### Comment · Reviewer_fnio · 2025-11-25
> >
> > Thank you for the response regarding both intuitive and theoretical explanations of the results, and clarification on Figure 5. I'd appreciate if you could add those intuitive notes in the paper to make the readers follow your ideas and motivations better.

---

> > > ### Author Response · Authors · 2025-11-26
> > > **Response to Reviewer 4 (fnio)**
> > >
> > > Thank you for your response. We are glad to hear that our clarifications addressed your concerns.
> > > We have added a brief theoretical pointer to Section 2.2 (Scaling Laws of LLMs).
> > > Regarding the intuitive explanations, although the main analysis already appears in Section 3.4 (Tokens per Parameter), we additionally incorporated an intuitive explanation into Section 4 (Discussion and Limitations) to make the motivation clearer for readers.
> > > Thank you again for your helpful suggestion.

---

### Official Review · Reviewer_tKtK · 2025-11-01

**Soundness:** 3
**Presentation:** 2
**Contribution:** 2
**Rating:** 6
**Confidence:** 3

**Summary:**

The paper empirically studies the effect of sparsity in sparse mixture of experts (MoEs) on downstream tasks. Specifically the paper trains a series of models based on Mixtral architecture and studies downstream performance on tasks that tend to rely on memorization vs tasks that test reasoning capabilities of language models (LMs). The paper studies the relationship between training loss and task loss and provides insights on the effect of active parameter count and tokens per parameter (TPP) on downstream tasks described above. The main findings include active FLOPs may lead to higher reasoning accuracy and that higher tokens per parameter may be preferable for reasoning tasks.

**Strengths:**

- The paper studies the effect of sparsity in MoEs on downstream tasks. This area has not been examined in detail so the study in the paper is timely and will likely be of interest to many researchers & practitioners.

- The empirical setup including models, data and downstream tasks are  described clearly in the paper. This gives me confidence that the experiments are reproducible.

- The models considered in the paper are not necessarily compute-optimal. This detail may provide additional insight into how to optimally train and use MoEs at scale.

- The paper proposes tokens per parameter (TPP) as a metric to track in addition to active parameters. This metric provides additional insight into the role of data in training MoEs that work well on downstream reasoning-type tasks. This may be interesting to many readers (and is definitely interesting to this reader)

**Weaknesses:**

- The paper considers a single architecture inspired by Mixtral family of MoEs in the work. It's understandable why this choice was made (experiment volume) but I do wonder if other architecture choices can change the conclusions made here. If possible, please discuss why Mixtral was chosen as opposed to other choices.

- The fact that memorization depends on total parameter count is known from prior literature. Furthermore, active number of parameters (inference FLOPs) have also observed to improve certain downstream tasks' performance (Abnar et al., 2205). So the claim that these are new contributions is weak.

- While sparsity in mentioned in the paper, the plots (Figure 1) for instance do not show this value but instead show top-K. This makes it hard for the reader to infer the effect of sparsity on empirical observations.

**Questions:**

- Is there a way to show how downstream performs with sparsity where sparsity is defined in the paper as 1 - (active / total experts)? Sparsity is mentioned in the paper but is not shown explicitly in scaling plots. Please include sparsity value, if possible, with the plots. Only Figure 5 appears to include sparsity (via density term which is its complement).

- The range of accuracy/error rate for GSM8K task appears to be on the lower side? Are these values good enough for readers to draw valid conclusions? A discussion on what is reasonable would be very useful to help the reader.

---

> ### Author Response · Authors · 2025-11-24
> **Response to Reviewer 3 (tKtK) - Part 1**
>
> We thank the reviewer for their thoughtful and constructive feedback.
>
> ## Architecture Choice
>
> > The paper considers a single architecture inspired by Mixtral family of MoEs in the work. It's understandable why this choice was made (experiment volume) but I do wonder if other architecture choices can change the conclusions made here. If possible, please discuss why Mixtral was chosen as opposed to other choices.
>
> Our study does not aim to exhaustively explore the full architectural design space of MoE models, as the number of possible variants including routing mechanisms, shared experts, and attention modifications is extremely large. Fully sweeping this space would be computationally infeasible.
>
> We adopted a Mixtral-style architecture. Mixtral maintains close structural compatibility with the widely used dense baseline Llama. Given the scale of our experiments, we could not afford to repeat the entire study across multiple architectures; therefore, we chose Mixtral so that a clean and controlled comparison between MoE and dense models would remain possible.
>
> Mixtral is also architecturally very close to recent state-of-the-art MoE models such as Qwen3 MoE [1], differing mainly in the use of QK-norm. In our regime, we did not observe the training instabilities that QK-norm is typically designed to mitigate. Similarly, OLMoE adopts a nearly identical MoE structure, with the primary architectural differences also concentrated around QK-norm related modifications.
>
> We excluded shared experts. Prior work OLMoE [2] reports negative effects from shared experts, and our own preliminary experiments indicated no meaningful differences once active and total FLOPs were matched. Including shared experts would therefore complicate interpretation without adding insight into sparsity, which is the primary focus of this study. For auxiliary terms such as load-balancing and router z-losses, we follow the OLMoE configuration to ensure stability while keeping the architectural core aligned with Mixtral.
>
> We have added this clarification to Section 4 (Discussion and Limitations) in the revised manuscript.
>
> [1] Qwen Team. Qwen3 Technical Report. 2025.
>
> [2] Muennighoff et al. OLMoE: Open Mixture-of-Experts Language Models. ICLR '25.
>
> ## Contributions
>
> > The fact that memorization depends on total parameter count is known from prior literature. Furthermore, active number of parameters (inference FLOPs) have also observed to improve certain downstream tasks' performance (Abnar et al., 2205). So the claim that these are new contributions is weak.
>
> We appreciate the reviewer’s thoughtful comment and agree that both the dependence of memorization on total parameter count and the benefit of increased active parameters for certain downstream tasks have been reported in prior work, including Abnar et al. (2025). Our work builds on these important observations. However, our contribution differs from prior literature in two key ways.
>
> First, most previous studies including Abnar et al. examine sparsity primarily through the lens of pre-training loss scaling, or show that sparsity-induced deficits can be mitigated by inference-time techniques such as Chain-of-Thought prompting. In contrast, our focus is on direct end-task reasoning accuracy. Through test-time compute scaling and GRPO, we show that even when additional inference compute is provided, the U-shaped trend at high sparsity persists. This indicates that the effect arises during pre-training and cannot be fully compensated by downstream compute an aspect not captured by pre-training-loss-based analyses.
>
> Second, we identify a previously unreported inverted-U relationship between sparsity and reasoning performance. Our findings suggest that this non-monotonic behavior arises because reasoning requires both sufficient active compute (active FLOPs per token) and an appropriate tokens-per-parameter (TPP) ratio.
>
> We believe our findings offer new empirical insight into the structure of sparsity.

---

> > ### Author Response · Authors · 2025-11-24
> > **Response to Reviewer 3 (tKtK) - Part 2**
> >
> > ## Clarification on Sparsity Visualization
> >
> > > While sparsity in mentioned in the paper, the plots (Figure 1) for instance do not show this value but instead show top-K. This makes it hard for the reader to infer the effect of sparsity on empirical observations.
> >
> > Thank you for pointing this out. We agree that sparsity is an important lens for interpreting our results, and we strive in the paper to present the phenomena from multiple complementary viewpoints (e.g., training loss, validation loss, accuracy, and model size) rather than relying on a single axis.
> >
> > We believe that Figure 5 already provides a clear and explicit visualization of sparsity. Its x-axis directly represents the ratio k/E, which is exactly the complement of sparsity and spans a broad range of regimes.
> >
> > Regarding Figure 1, the x-axis already corresponds to sparsity in a monotonic way: within each fixed top-k group, increasing the number of experts increases the total parameter count and therefore increases sparsity. For the two task-loss plots on the right, re-expressing the x-axis directly in terms of sparsity would yield curves that behave identically to the corresponding trends shown in Figure 5.
> >
> > To make the correspondence clearer to readers, we have updated the caption of Figure 1 to note that moving right within each top-k group corresponds to increasing sparsity.
> >
> > ## Clarification on Figure 5
> >
> > > Is there a way to show how downstream performs with sparsity where sparsity is defined in the paper as 1 - (active / total experts)? Sparsity is mentioned in the paper but is not shown explicitly in scaling plots. Please include sparsity value, if possible, with the plots. Only Figure 5 appears to include sparsity (via density term which is its complement).
> >
> > Thank you for raising this point. We would like to clarify that Figure 5 already visualizes sparsity.
> > The x-axis explicitly represents the ratio of active experts $k$ to total experts $E$, which is simply the complement of sparsity.
> > To make this clearer, configurations such as $k=2, E=256$ appear in the plot at $k/E = 1/128$. Likewise, settings such as $k=8, E=128$ or $k=4, E=64$ both map to $k/E = 1/16$.
> > These examples illustrate that each point on the x-axis aggregates multiple model configurations that share the same sparsity level, and that the plot already spans a broad range of sparsity regimes.
> > In response to your suggestion, and consistent with related feedback from Reviewer pkQZ, we have updated the figure caption to explicitly state that the x-axis denotes “the ratio of active experts $k$ to total experts $E$,” so that the connection to sparsity is immediately clear to readers.
> >
> > ## On the Magnitude and Reliability of GSM-Plus Performance Differences
> >
> > > The range of accuracy/error rate for GSM8K task appears to be on the lower side? Are these values good enough for readers to draw valid conclusions? A discussion on what is reasonable would be very useful to help the reader.
> >
> > We would like to clarify that we evaluated our models on GSM-Plus, which contains 10,552 instances. For our models, the average standard error of accuracy on GSM-Plus is approximately 0.0030 percentage points, and the observed performance differences of around 0.0083 percentage points are therefore larger than the corresponding standard error. This indicates that the differences we report are statistically meaningful.

---

> > > ### Comment · Reviewer_tKtK · 2025-11-26
> > >
> > > I acknowledge the rebuttal and more importantly commend the authors for their thoughtful response to my questions. In general, I am satisfied with the responses. I encourage the authors to find a way to highlight the "U-shaped" observation in the abstract itself but this is not a requirement. Might just help readers (including this reviewer) appreciate the finding made in this work.
> > >
> > > This is a good paper.

---

### Official Review · Reviewer_pkQZ · 2025-11-01

**Soundness:** 3
**Presentation:** 4
**Contribution:** 3
**Rating:** 6
**Confidence:** 3

**Summary:**

This paper aims to investigate how MoE sparsity influences two distinct capability regimes: memorization skills and reasoning skills. The work shows how Active FLOP is more important for reasoning, while memorization improves with number of total parameters. Another interesting finding provided in this work is that changing the k in top-k routing has a negligible effect if the number of active parameters is kept constant.

**Strengths:**

1. It is an important observation that for MoE models, downstream accuracy can deviate from the predictions of conventional scaling laws, and these deviations may vary across different tasks.

2. Exhaustive experimentation is done in reasoning and coding tasks to demonstrate the U shape of tasks performance with the increase of total parameters at a FLOP controlled setting

3. Exhaustive experiments are done to show that post training couldn't improve this.

**Weaknesses:**

1. The number of tokens used seems small to if we are targeting End task performance, specially for MOE models
2. It would be good to get some ablation for various router choices, though than can be a future work
3. In Page 9, figure 8, it would be good do the study at k>1 (ideally 8) and E >8
4. More details about the post training setup is helpful. How many tokens in the post training set?
5. No details have been provided whether Continuous training is done or learning rate is annealed before evaluating end task

**Questions:**

1. The number of tokens used seems small to if we are targeting End task performance, specially for MOE models
2. It would be good to get some ablation for various router choices, though than can be a future work
3. In Page 9, figure 8, it would be good do the study at k>1 (ideally 8) and E >8
4. More details about the post training setup is helpful. How many tokens in the post training set?
5. No details have been provided whether Continuous training is done or learning rate is annealed before evaluating end task

---

> ### Author Response · Authors · 2025-11-24
> **Response to Reviewer 2 (pkQZ) - Part 1**
>
> We sincerely thank the reviewer for their constructive feedback and for recognizing the significance of our findings regarding the deviation from conventional scaling laws.
>
> ## 1. Pre-training Token
>
> > The number of tokens used seems small to if we are targeting End task performance, specially for MOE models
>
> We acknowledge the reviewer's concern regarding the token budget (125B) relative to end-task performance. While we agree that MoE models generally benefit from large-scale data, we selected this budget based on two primary factors: computational constraints relative to scaling laws, and the scarcity of high-quality reasoning data available in the open datasets.
>
> 1. Computational Cost and Scale Comparison: Although 125B tokens may appear small compared to frontier model training, the computational cost for our experimental sweep is substantial.
> Our largest single run (width=2048, top-16) consumes approximately $5.5 \times 10^{21}$ FLOPs.
> This scale exceeds the experimental budgets found in recent scaling law literature.
> For instance, [1] explore up to $1 \times 10^{21}$ FLOPs, and our scale is comparable to the iso-FLOPs regimes discussed in [2] ($3 \times 10^{21}$ FLOPs).
>
> 2. Our preliminary ablations indicated that simply scaling the token count yields diminishing returns for reasoning tasks. We attribute this primarily to the scarcity of high-quality reasoning and mathematics datasets available in the open datasets at the time of this study.
> We have added a detailed ablation study to Appendix C.8, comparing our 125B baseline against a model trained on 1T tokens. As shown in the table below (and Table 4 in the revision), while memorization tasks (TriviaQA) saw massive gains due to the rich world knowledge in web text, reasoning tasks (GSM8K, GSM-Plus) showed comparatively marginal improvements. This confirms that under the constraint of available open-source data, simply scaling the volume of general web text contributes minimally to reasoning capabilities, justifying the use of a 125B budget to prioritize architectural exploration.
>
> | Token Budget | TriviaQA | HellaSwag | GSM8K | GSM-Plus |
> | :--- | :--- | :--- | :--- | :--- |
> | 125B | 0.279 | 0.426 | 0.318 | 0.188 |
> | 1T | 0.535 | 0.538 | 0.363 | 0.214 |
>
> [1] Abnar et al. Parameters vs flops: Scaling laws for optimal sparsity for mixture-of-experts language models. ICML ‘25.
>
> [2] Hoffmann et al. An empirical analysis of compute-optimal large language model training. NeurIPS ‘22.
>
>
> ## 2. Scope Regarding Router Ablations
>
> > It would be good to get some ablation for various router choices, though than can be a future work
>
> We thank the reviewer for the suggestion and agree that ablating different routing mechanisms would provide valuable insights into the robustness of the observed scaling trends. However, given the substantial computational scale of our experiments, performing a systematic sweep over router architectures was infeasible within the scope of this study. We explicitly acknowledge that investigating how alternative routing strategies influence the sparsity-reasoning trade-off is an intriguing direction for future work.

---

> > ### Author Response · Authors · 2025-11-24
> > **Response to Reviewer 2 (pkQZ) - Part 2**
> >
> > ## 3. Clarification on Experimental Regimes
> >
> > > In Page 9, figure 8, it would be good do the study at k>1 (ideally 8) and E >8
> >
> > We would like to clarify that our experiments already cover the regimes of $k>1$ and $E>=8$.
> > In our plots analyzing the "ratio of active experts $k$ to total experts $E$", we aggregate models with the same sparsity ratio. For instance, the data point at the ratio $1/128$ includes the results for the configuration $k=2, E=256$. Similarly, the ratio $1/16$ aggregates multiple settings such as $k=8, E=128$ and $k=4, E=64$.
> > To prevent ambiguity, we have updated the figure captions to explicitly define the x-axis as "the ratio of active experts $k$ to total experts $E$".
> >
> > ## 4. Clarification on Post-Training Setup
> >
> > > More details about the post training setup is helpful. How many tokens in the post training set?
> >
> > We thank the reviewer for pointing this out. Although the post-training details were included in Appendix A.2 of the original submission, we realized that a clear reference to this section was missing from the main text.
> > We have updated the manuscript to explicitly state: "Further details regarding the training setup and hyperparameters are provided in Appendix A.2."
> >
> > We used a global batch size of 1,024 and trained for 15 epochs, which corresponds to 105 training steps. This amounts to a total of 107,520 samples processed during post-training.
> >
> > ## 5. Clarification on Learning Rate Schedule
> >
> > > No details have been provided whether Continuous training is done or learning rate is annealed before evaluating end task
> >
> > We would like to clarify that this information was in the submitted manuscript.
> > Regarding the learning rate schedule, we explicitly described the process in Section 3.1 EXPERIMENTAL SETUP. As written in the paper:
> > "We train with AdamW (Loshchilov & Hutter, 2019) using a peak learning rate of $4\times10^{-4}$, a 2k-step linear warm-up followed by cosine decay, and a weight decay of 0.1."
> >
> > Additionally, the details of the corpus used for this pre-training are provided in the appendix, as noted in Table 1: Breakdown of the 125 B-token pre-training corpus.
> >
> > We hope this clarification addresses your concerns.

---

### Official Review · Reviewer_p7WF · 2025-11-02

**Soundness:** 3
**Presentation:** 3
**Contribution:** 3
**Rating:** 6
**Confidence:** 3

**Summary:**

The paper studies the optimal sparsity of Mixture-of-Experts models under memorization and reasoning skills by training MoE models with varying total parameters, sparsity, and top-k routing under the fixed budget. Through extensive experiments, the paper concludes that 1. the downstream reasoning quality is decided by both the active FLOPs and pretraining loss and 2. there exist different optimal tokens-per-parameter ratios for memorization and reasoning tasks.

**Strengths:**

- The paper is well-written and easy to understand.
- The experiments are comprehensive while supporting the major claims of the paper.
- One of the main findings is surprising, as it shows that higher sparsity only improves performance under memorization instead of reasoning tasks under the iso-FLOP settings.

**Weaknesses:**

- The paper might need to address more about its intuition and originality from previous works such as [1] and [2], since similar observations regarding the optimal sparsity in MoE models have been made.
- Theoretical insights are encouraged to explain the experimental findings.
- The U-shape trend plot for reasoning tasks in Figure 2 is very interesting, and I suggest the authors to verify such finding under more reasoning tasks.

[1] Samira, Abnar, et al. "Parameters vs FLOPs: Scaling Laws for Optimal Sparsity for Mixture-of-Experts Language Models." arXiv:2501.12370 (2025).

[2] Zhao, Jinze, et al. "Sparse Mixture-of-Experts for Compositional Generalization: Empirical Evidence and Theoretical Foundations of Optimal Sparsity." arXiv:2410.13964 (2025)

**Questions:**

Questions are addressed in the Weaknesses section.

---

> ### Author Response · Authors · 2025-11-24
> **Response to Reviewer 1 (p7WF)**
>
> We thank the reviewer for their insightful comments.
>
> ## Originality and intuition
>
> > The paper might need to address more about its intuition and originality from previous works such as [1] and [2], since similar observations regarding the optimal sparsity in MoE models have been made.
>
> We thank the reviewer for raising the question regarding our intuition and originality relative to prior works. As Reviewer tKtK noted, the effect of MoE sparsity on downstream reasoning has not been examined in detail in previous studies. We appreciate the opportunity to clarify our distinct contribution.
>
> Our key contribution is the discovery of an inverted-U for reasoning-optimal sparsity. We attribute this to a fundamental tension between
>
>  (1) Active FLOPs, Increasing active parameters benefits computation for reasoning.
>
> (2) Tokens-per-parameter (TPP), where increasing sparsity lowers the effective TPP and moves the model away from the optimal data–total-parameter balance for reasoning.
>
>
> Reasoning peaks only when both are balanced.
>
> ### Relation to [1].
>
> Abnar et al. study sparsity through pre-training loss scaling and argue that sparsity-induced deficits can be mitigated via inference-time techniques such as CoT.
> In contrast, we focus directly on reasoning accuracy and show through test-time compute scaling and GRPO that, the U-shaped trend persists and the relative degradation at high sparsity remains unchanged.
>
> ### Relation to [2].
>
> Zhao et al. show that compositional tasks benefit from larger Top-k.
> Our findings differ: in math and code reasoning tasks, increasing Top-k alone is insufficient. The interaction with TPP produces a non-monotonic curve absent in [2].
>
> ## Theoretical Perspective
>
> > Theoretical insights are encouraged to explain the experimental findings.
>
> Our work is primarily empirical, but prior work summarized in Section 2.2, particularly Jelassi et al. (2025) provide the most relevant theoretical intuition for interpreting our results. They show that, in Graph Neural Networks, sparse MoE models can solve memorization-heavy tasks more efficiently than dense models, whereas excessive sparsity might decrease the performance of reasoning-intensive tasks such as the Length-2 Path Problem. While our tasks, math and coding, differ from those tasks in GNNs, their theoretical results offer a useful intuition: the optimal level of sparsity differs between reasoning tasks and memorization-oriented tasks. This theoretical perspective aligns with and helps to understand our empirical results.
>
> ## Additional Verification of the U-Shape Trend
>
> > The U-shape trend plot for reasoning tasks in Figure 2 is very interesting, and I suggest the authors to verify such finding under more reasoning tasks.
>
> Our submitted manuscript already demonstrates the inverted U-shaped trend across four reasoning benchmarks: GSM8K, GSM-Plus, HumanEval, and MBPP.
>
> To further address the reviewer’s suggestion, we attempted to evaluate additional challenging tasks such as GPQA and AIME. However, for the model scales considered, performance was indistinguishable from random guessing, making it impossible to extract reliable scaling signals.
>
> Instead, we added HumanEval+ and MBPP+ to expand the coverage while remaining within a stable performance regime. Both benchmarks exhibit the same inverted U-shaped trend, reinforcing the robustness of our findings. We have included these results in the revised Appendix (Figure 22).

---

### Meta-Review · Area_Chair_fF6N · 2026-01-04

**Summary:**

The paper addresses a critical gap in the understanding of scaling laws for LLMs, specifically regarding Mixture-of-Experts (MoE) architectures. While MoE models are standard in frontier systems, the authors convincingly demonstrate that the "compute-optimal" sparsity differs significantly between memorization and reasoning tasks. The paper establishes that while memorization skills scale monotonically with parameter count (sparsity), reasoning skills exhibit a non-monotonic, inverted U-shaped relationship with Tokens Per Parameter (TPP). The study uncovers an important trade-off between memorization and reasoning with TPP, where memorization skills are parameter-hungry and reasoning skills are data-hungry. Besides, the authors show that models with identical pre-training loss can have vastly different reasoning capabilities, determined largely by Active FLOPs (controlled by top-k routing). This disentanglement of loss and downstream accuracy is a valuable contribution to model design. This is a unamimous acceptance paper.

**Reviewer Concerns:**

There are no particularly interesting concerns. Most concerns are just misunderstandings or non-essential. After rebuttal, I think only one valid concern remains: Scalability to "Hard" Reasoning (Reviewer p7WF). The authors admitted that they could not verify their trends on harder reasoning benchmarks like GPQA or AIME because their model scale was too small to achieve non-random performance. Thus, whether the "U-shaped" sparsity trend holds for advanced reasoning capabilities remains an extrapolation.

**Reviewer Scores:**

I believe reviewers' current scores are valid and will remain unchanged.

---

### Decision · Program_Chairs · 2026-01-26

Accept (Oral)